EMBO
Molecular Medicine

# SARS-CoV-2 spike protein enhances MAP4K3/GLK-induced ACE2 stability in COVID-19

Huai-Chia Chuang[1,*] , Chia-Hsin Hsueh[1] , Pu-Ming Hsu[1] , Rou-Huei Huang[1], Ching-Yi Tsai[1],
Nai-Hsiang Chung[2,3] , Yen-Hung Chow[2,4,**] & Tse-Hua Tan[1,5,***]

## Abstract

ACE2 on epithelial cells is the SARS-CoV-2 entry receptor. Single-cell RNA-sequencing data derived from two COVID-19 cohorts revealed that MAP4K3/GLK-positive epithelial cells were increased in patients. SARS-CoV-2-induced GLK overexpression in epithelial cells was correlated with COVID-19 severity and vesicle secretion. GLK overexpression induced the epithelial cell-derived exosomes containing ACE2; the GLK-induced exosomes transported ACE2 proteins to recipient cells, facilitating pseudovirus infection. Consistently, ACE2 proteins were increased in the serum exosomes from another COVID-19 cohort. Remarkably, SARS-CoV-2 spike protein-stimulated GLK, and GLK stabilized ACE2 in epithelial cells. Mechanistically, GLK phosphorylated ACE2 at two serine residues (Ser776, Ser783), leading to the dissociation of ACE2 from its E3 ligase UBR4. Reduction in UBR4-induced Lys48-linked ubiquitination at three lysine residues (Lys26, Lys112, Lys114) of ACE2 prevented its degradation. Furthermore, SARS-CoV-2 pseudovirus or live virus infection in humanized ACE2 mice induced GLK and ACE2 protein levels, and ACE2-containing exosomes. Collectively, ACE2 stabilization by SARS-CoV-2-induced MAP4K3/GLK may contribute to the pathogenesis of COVID-19.

**Keywords** MAP4K3/GLK; ACE2; UBR4; SARS-CoV-2; COVID-19
**Subject Categories** Microbiology, Virology & Host Pathogen Interaction; Proteomics
**EMBO Mol Med (2022) e15904**

## Introduction

SARS-CoV-2 infection in human induces the coronavirus disease 2019 (COVID-19). SARS-CoV-2 enters epithelial cells through the receptor angiotensin-converting enzyme 2 (ACE2; Wan *et al*, 2020), which is mainly expressed in epithelial cells of the airways and lungs (Salamanna *et al*, 2020). ACE2 mRNA is also expressed in multiple tissues, including the kidneys, liver, colon, heart, retina, and ileum (Sungnak *et al*, 2020). Although ACE2 mRNA levels are generally low, single-cell RNA-sequencing (scRNA-seq) data using human nasal swab samples indicate that the numbers of ACE2-positive epithelial cells are increased in COVID-19 patients compared with healthy controls (Chua *et al*, 2020). It is likely that ACE2 mRNA or protein induction in epithelial cells may play a crucial role in COVID-19 pathogenesis.

The serine/threonine kinase MAP4K3 (also named GLK) induces NF-κB activation and IL-17A production (Chuang *et al*, 2011, 2018). NF-κB activation induces transcription of multiple proinflammatory cytokines; inhibition of GLK results in suppression of NF-κB–mediated cytokine production (Chuang *et al*, 2019b). In addition, GLK overexpression in T cells induces IL-17A production and T-cell hyperactivation, leading to autoimmune diseases (Chen *et al*, 2012, 2013; Chuang *et al*, 2018). Moreover, GLK overexpression in lung epithelial cells is correlated with human lung cancer recurrence and poor prognosis (Hsu *et al*, 2016). GLK is also overexpressed in human liver cancer (Ho *et al*, 2016). GLK directly phosphorylates the cytoskeleton regulator IQGAP1, contributing to the enhancement of cell migration and cancer metastasis (Chuang & Tan, 2019; Chuang *et al*, 2019a). In this report, we explored whether GLK is involved in COVID-19 pathogenesis by scRNA-seq and proteomics analyses using four COVID-19 patient cohorts, including one newly enrolled COVID-19 patient cohort from Taiwan. Our results suggest that GLK stabilizes ACE2 proteins in epithelial cells, enhancing SARS-CoV-2 infection susceptibility of epithelial cells.

## Results

### MAP4K3/GLK is overexpressed in epithelial cells of COVID-19 patients

We analyzed GLK levels using published single-cell RNA-sequencing (scRNA-seq) data (GSE145926; Cohort #1) of

---

1  Immunology Research Center, National Health Research Institutes, Zhunan, Taiwan
2  National Institute of Infectious Disease and Vaccinology, National Health Research Institutes, Zhunan, Taiwan
3  Institute of Molecular and Cellular Biology, National Tsing Hua University, Hsinchu, Taiwan
4  Graduate Institute of Biomedical Sciences, China Medical University, Taichung, Taiwan
5  Department of Pathology & Immunology, Baylor College of Medicine, Houston, TX, USA
   *Corresponding author. Tel: +886 37 206 166 ext. 37612; E-mail: cinth@nhri.edu.tw
   **Corresponding author. Tel: +886 37 206 166 ext. 37738; E-mail: choeyenh@nhri.edu.tw
   ***Corresponding author. Tel: +886 37 206 166 ext. 37601; E-mail: ttan@nhri.edu.tw

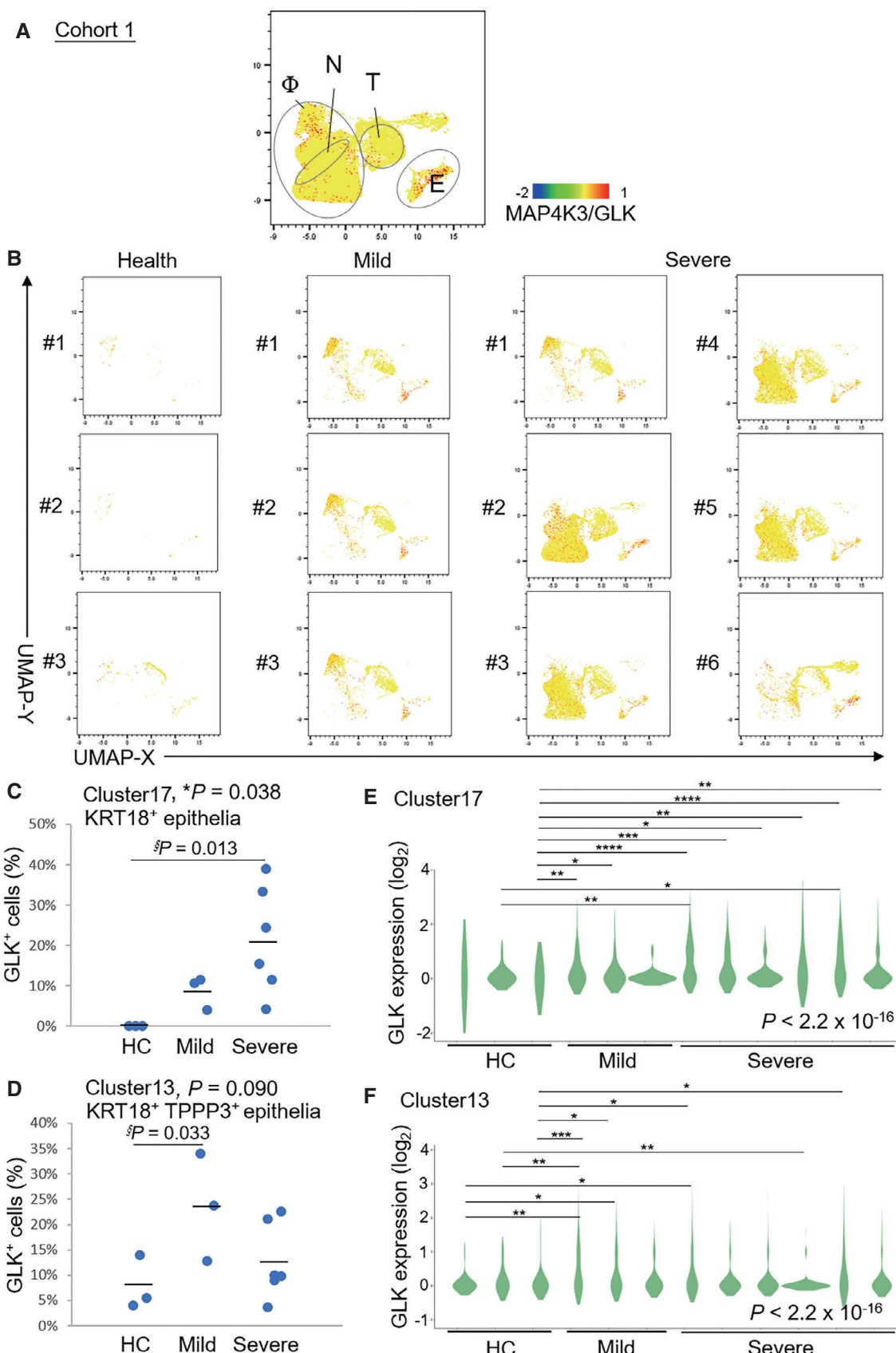

**Figure 1.**

**Figure 1.  Induction of MAP4K3/GLK in epithelial cells of BALFs from COVID-19 patients.**

A, B  Distribution and classification of MAP4K3/GLK-positive cells in 20 ml BALFs of all 12 individuals (A), three healthy controls (HC, #1–#3), three mild COVID-19 patients (#1–#3), and six severe COVID-19 patients (#1–#6) (B) from Cohort #1. Data were shown in UMAP. Single-cell gene expression of MAP4K3 (GLK) in individual cells was shown in color scale. Φ denotes macrophages; N denotes neutrophils; T denotes T cells; E denotes epithelial cells.

C, D  The percentages of MAP4K3/GLK-positive KRT18$^+$ epithelial (C) or KRT18$^+$ TPPP3$^+$-ciliated epithelial cells (D) in BALFs from Cohort #1.

E, F  GLK mRNA levels in KRT18$^+$ epithelial (E) or KRT18$^+$ TPPP3$^+$-ciliated epithelial cells (F) of Cohort #1. $n$ (cell number) = 3, 14, 33 for HC; $n$ = 65, 43, 74 for mild COVID-19 patients; $n$ = 164, 62, 10, 4, 74, 66 for severe COVID-19 patients (E). $n$ = 25, 43, 36 for HC; $n$ = 209, 244, 141 for mild COVID-19 patients; $n$ = 189, 223, 330, 792, 80, 243 for severe COVID-19 patients (F).

Data information: In (C, D), *, *P*-value < 0.05 (ANOVA test); §, *P*-value < 0.05 (Dunnett's test). In (E, F), **P*-value < 0.05; ***P*-value < 0.01; ****P*-value < 0.001; *****P*-value < 0.0001 (Kruskal–Wallis test).

---

bronchoalveolar lavage fluid (BALF) cells from nine COVID-19 patients (six severe and three moderate) and three healthy controls (Liao *et al*, 2020). Consistent with the published report (Liao *et al*, 2020), macrophage, neutrophil, T cell, B cell, NK cell and epithelial cell, but not dendritic cell (DC), populations were increased in severe COVID-19 patients compared with healthy controls (Appendix Fig S1). Interestingly, GLK expression was highly induced in epithelial cells and macrophages in BALFs from COVID-19 patients (Fig 1A and B). Strikingly, the percentage of GLK-positive KRT18$^+$ epithelial cells in BALF cells was drastically increased in COVID-19 patients (Fig 1C). Similarly, the percentage of GLK-positive KRT18$^+$ TPPP3$^+$-ciliated epithelial cells were also induced in COVID-19 patients (Fig 1D). Moreover, the GLK levels were increased in BALF epithelial cells of COVID-19 patients compared to those of healthy controls (Fig 1E and F). To validate the findings using COVID-19 BALF cells, we also analyzed GLK-positive epithelial cells of another published scRNA-seq data (FigShare 12436517; Cohort #2) using nasal swab cells of 19 COVID-19 patients (11 severe and 8 moderate) and five healthy controls (Chua *et al*, 2020; Appendix Fig S2). Consistently, the percentage of GLK-positive epithelial cells were also increased in nasal swab samples from Cohort #2 COVID-19 patients compared with healthy controls (Appendix Fig S3A and B). The GLK levels were also increased in epithelial cells of nasal swab samples from Cohort #2 COVID-19 patients (Appendix Fig S3C and D). To test whether SARS-CoV-2 infection results in GLK overexpression in lung epithelial cells, HCC827 lung epithelial cells were infected with SARS-CoV-2 pseudovirus. We found that both protein levels and mRNA levels of GLK were increased in SARS-CoV-2 pseudovirus-infected HCC827 lung epithelial cells (Fig 2A and B), whereas GLK levels were not induced by vesicular stomatitis virus-G (VSV-G) pseudotyped lentivirus (Appendix Fig S4). GLK levels were also increased by the spike protein of SARS-CoV-2 (Fig 2C and D). Collectively, these results suggest that SARS-CoV-2 infection induces GLK overexpression in lung epithelial cells of COVID-19 patients.

## GLK-induced exosomes transport ACE2 to other epithelial cells, enhancing SARS-CoV-2 pseudovirus infection

To understand the biological function of GLK overexpression in epithelial cells during SARS-CoV-2 infection, we analyzed the enriched pathways of GLK-overexpressing epithelial cells using KEGG (Kyoto Encyclopedia of Genes and Genomes) pathway analysis. The KEGG pathway analysis using scRNA-seq data of the epithelial cells from both Cohort #1 (Liao *et al*, 2020) and Cohort #2 (Chua *et al*, 2020) revealed that the upregulated genes in GLK-positive epithelial cells mainly belong to the pathways of vesicle, intracellular vesicle, extracellular vesicle, and viral process (Fig 2E–H).

Virus-induced extracellular vesicle release provides a supportive function for virus infection (Hassanpour *et al*, 2020). We next tested whether GLK induces vesicle/exosome production from epithelial cells. Interestingly, the number of epithelial cell-derived exosomes (extracellular vesicle diameter < 200 nm) was drastically increased by overexpression of GLK wild-type but not GLK kinase-dead (K45E) mutant (Fig 3A). Remarkably, ACE2 proteins levels were increased in the exosomes isolated from GLK wild-type-, but not GLK kinase-dead mutant-, overexpressing HCC827 lung epithelial cancer cells by immunoblotting analysis (Fig 3B). The immunoblotting data displayed two major bands of endogenous ACE2 proteins (Fig 3B). The higher molecular-weight band (about 130 kDa) is glycosylated ACE2, while the lower molecular-weight band is nonglycosylated ACE2 (Tipnis *et al*, 2000; Li *et al*, 2003). In contrast, the transfected ACE2 proteins were detected as one major band (about 130 kDa; see below). To study whether ACE2 proteins are present in exosomes of COVID-19 patients, we analyzed ACE2 proteins in serum exosomes from seven enrolled human COVID-19 patients and four enrolled healthy controls from NHRI Biobank for COVID-19 patients in Taiwan (Cohort #3) by immunoblotting analyses. ACE2 protein levels were increased in the serum exosomes isolated from COVID-19 patients compared with healthy controls (Fig 3C). The increased levels of the exosome marker CD63 in serum exosomes

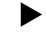

**Figure 2.  The upregulated genes in GLK-positive epithelial cells belong to the pathways of intracellular/extracellular vesicle and viral process.**

A–D  Endogenous GLK levels were increased by the SARS-CoV-2 spike protein. Immunoblotting analysis of the endogenous GLK and vinculin proteins from the lysates of HCC827 lung epithelial cells either infected with SARS-CoV-2 pseudovirus (A) or treated with the SARS-CoV-2 spike protein (C). Real-time PCR analysis of mouse GLK mRNA levels in HCC827 lung epithelial cells either infected with SARS-CoV-2 pseudovirus (B) or treated with the SARS-CoV-2 spike protein (D). The mRNA levels of GLK were normalized to GAPDH mRNA levels. $n$ = 2 (technical replicates) per group.

E–H  KEGG (Kyoto Encyclopedia of Genes and Genomes)-enriched pathways of the upregulated genes in GLK-positive KRT18$^+$ epithelial cells or KRT18$^+$ TPPP3$^+$-ciliated epithelial cells from Cohort #1 (E and F) and Cohort #2 (G and H). Pathways belonging to different classifications are listed on the left side of the plot. Varied numbers of genes enriched in individual pathways are presented by different diameter sizes and numbers for individual dots. Adjusted *P*-value is ranging from 0 ~ 1; less *P*-value means greater intensiveness.

Data information: In (B, D), data are presented as mean values. Data shown are representative results of three (A–D) independent experiments.
Source data are available online for this figure.

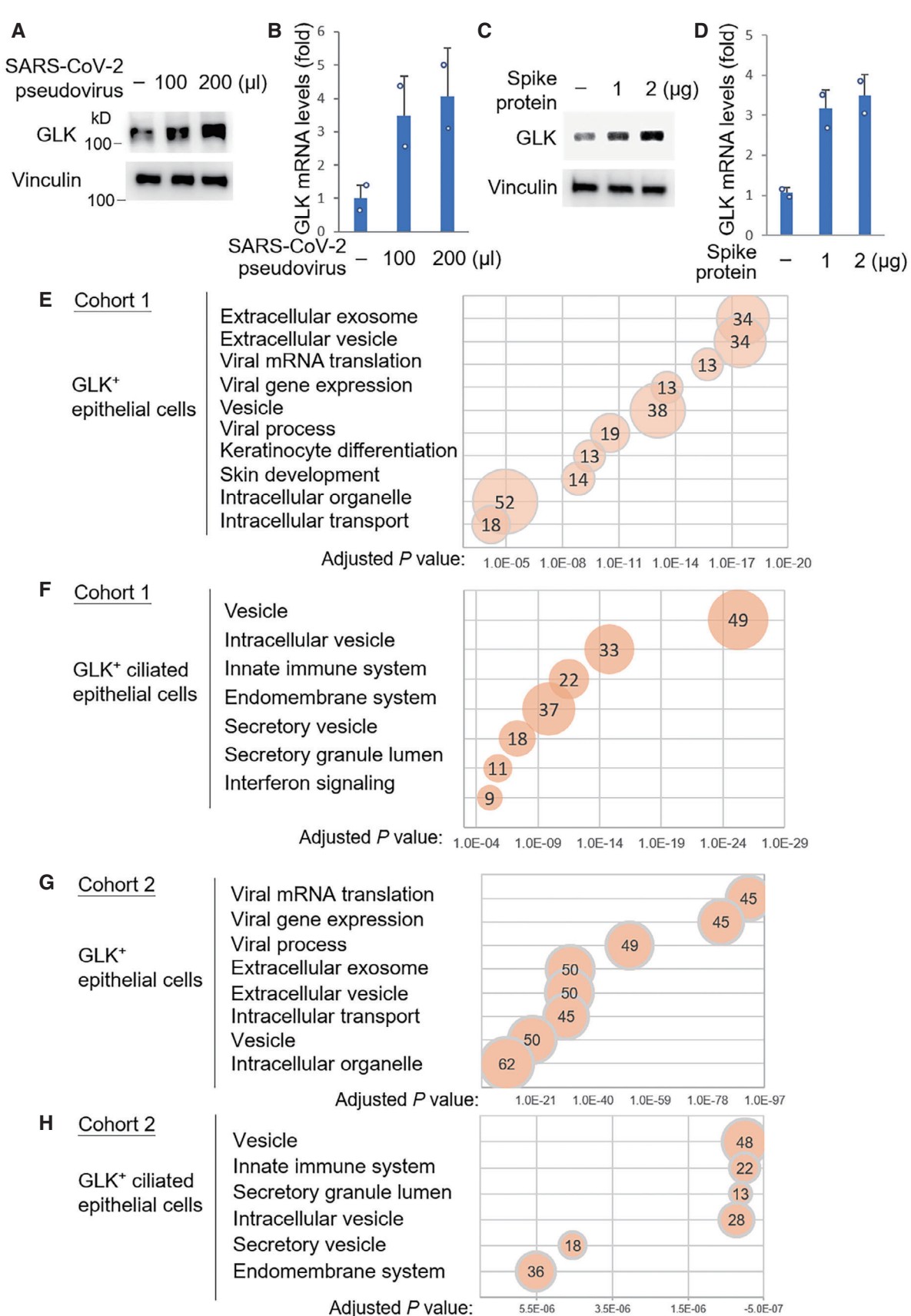

Figure 2.

suggest an enhancement of exosome release in COVID-19 patients (Fig 3C).

The serum exosomes from six additional COVID-19 patients were subjected to proteomics. Multiple serum exosomal proteins were identified from COVID-19 patients and healthy controls (Dataset EV1). The Cellular Component Ontology analysis shows that COVID-19 patient-enriched exosomal proteins mainly belong to membrane proteins, lysosomal proteins, and mitochondria proteins, while healthy control-enriched exosomal proteins mainly belong to cytosolic and nuclear proteins (Appendix Fig S5). Moreover, KEGG pathway analysis shows that COVID-19 patient-enriched exosomal proteins may be involved in multiple metabolic pathways (Appendix Fig S6). Notably, ACE2 protein peptides were detected in serum exosomes from all six COVID-19 patients by mass spectrometry analyses (Fig EV1). In contrast, ACE2 peptides were detected in serum exosomes from only two of six healthy controls (Fig EV1); furthermore, the protein scores of ACE2 in the serum exosomes of these two healthy controls were lower than those of COVID-19 patients (Fig EV1). The data also showed that the ACE2 peptides detected in the serum exosomes of COVID-19 patients matched the ACE2 protein sequences spanning from the N terminus to the C terminus of the ACE2 protein including the ectodomain (residues 1–740), the transmembrane domain (residues 741–762), and the cytoplasmic tail (residues 762–805; Fig EV1). Because soluble ACE2 proteins would not contain ACE2 transmembrane domain and cytoplasmic tail (residues 741–805; Lambert *et al*, 2005), the peptides of ACE2 transmembrane and cytoplasmic domains detected in the serum exosomes of COVID-19 patients are not soluble ACE2 proteins. To validate whether ACE2 proteins are also present in the sera of COVID-19 patients from another cohort, we analyzed mass spectrometry data of serum samples from the fourth human COVID-19 cohort (Cohort #4; Shen *et al*, 2020). As expected, ACE2 protein levels in the sera were significantly increased in COVID-19 patients from Cohort #4 (Shen *et al*, 2020; Appendix Fig S7A). Consistently, the ACE2 peptides detected in the sera of Cohort #4 COVID-19 patients also contained ACE2 transmembrane domain and

cytoplasmic tail (29 of 97 peptides; Appendix Table S1), supporting that the detected ACE2 peptides are derived from exosomal ACE2 proteins rather than soluble ACE2 proteins co-precipitated with serum exosomes. These results from both Cohort #3 and Cohort #4 suggest that ACE2 protein levels in the serum exosomes are increased in COVID-19 patients.

Exosome-transported ACE2 proteins from endothelial progenitor cells promote the survival and function of recipient endothelial cells (Wang *et al*, 2020). Next, we examined whether exosomes derived from GLK plus Flag-ACE2-overexpressing cells can be transported to co-incubated epithelial cells. Exogenous Flag-tagged ACE2 (in red) was detected within recipient epithelial cells that were incubated with the exosomes derived from the cells co-transfected with GLK and Flag-ACE2 (Fig 3D), whereas Flag-tagged ACE2 was not transported into recipient cells by the exosomes derived from Flag-ACE2-expressing cells without GLK overexpression (Fig 3D). GLK overexpression in epithelial cells does not cause cell damage/death (Chuang *et al*, 2019a). These results suggest that GLK overexpression in epithelial cells induces the release of exosomes and may facilitate the transport of ACE2 proteins to other cells via exosomes. To study whether the exosome-transported ACE2 proteins are functional receptors for the spike protein of SARS-CoV-2, we further incubated the exosome-recipient cells with SARS-CoV-2 spike (S) proteins. The S proteins (in green color) were shown in early endosomes (in red color) of the recipient cells that were incubated with exosomes derived from GLK-overexpressing cells but not GLK (K45E) kinase-dead mutant-overexpressing cells (Fig 3E). The confocal imaging data suggest that ACE2 proteins transported by GLK-induced exosomes may facilitate SARS-CoV-2 infection. To further support this finding, the exosome-recipient cells were treated with SARS-CoV-2 pseudovirus. Recipient epithelial cells that were incubated with exosomes derived from GLK-overexpressing epithelial cells were more efficiently infected with SARS-CoV-2 pseudovirus than those incubated with exosomes from GLK (K45E) kinase-dead mutant-overexpressing cells (Fig 3F). Collectively, these results suggest that GLK overexpression in lung epithelial cells induces the

---

**Figure 3.  MAP4K3/GLK induces ACE2-containing exosomes from epithelial cells.**

A  ZetaView nanoparticle tracking analysis of particle numbers and sizes of CD63[+] extracellular vesicles (EVs) isolated from the supernatants of HCC827 lung epithelial cancer cells, which were transfected with GLK wild-type or GLK kinase-dead (K45E) mutant. EVs were isolated sequentially using ExoQuick kits and then ExoQuick ULTRA columns.

B  Immunoblotting of ACE2 and CD63 proteins in exosomes isolated from GLK wild-type- or GLK (K45E) kinase-dead-overexpressing HCC827 cells. Exosomes were isolated sequentially using ExoQuick kits and then ExoQuick ULTRA columns. Arrowhead denotes glycosylated ACE2 proteins; asterisk denotes nonglycosylated ACE2 proteins.

C  Immunoblotting of ACE2 and CD63 proteins in serum exosomes isolated from the sera of four healthy controls (HC) and seven COVID-19 patients from NHRI Biobank for COVID-19 patients in Taiwan (Cohort #3). Serum collection days from the onset are shown; treatment of oxygen therapy on the patient is also indicated. Exosomes were isolated sequentially using ExoQuick kits and then ExoQuick ULTRA columns. Arrowhead denotes glycosylated ACE2 proteins; asterisk denotes nonglycosylated ACE2 proteins.

D  Confocal microscopy analysis of Flag-tagged ACE2 proteins (red) in recipient cells after incubation with exosomes for 72 h. Exosomes were isolated sequentially from the supernatants of Flag-ACE2-overexpressing or GFP-GLK plus Flag-ACE2-coexpressing HCC827 cells using ExoQuick kits and then ExoQuick ULTRA columns. Arrows denote Flag-tagged ACE2 proteins. Original magnification, ×630; scale bars, 10 μm.

E  Confocal microscopy analysis of SARS-CoV-2 spike (S) protein (green) and RFP-labeled early endosomes (red) in exosome-recipient HCC827 epithelial cells. After incubation with exosomes for 72 h, recipient cells were treated with the spike protein (S) for another 24 h. Yellow color suggests the localization of S protein in early endosomes. Original magnification, ×630; scale bars, 25 μm.

F  Cell entry efficiencies of SARS-CoV-2 pseudovirus into exosome-recipient HCC827 epithelial cells were measured by luciferase activity and presented as relative light units (RLU) at 24 h postinfection. *n* = 3 (biological replicates).

Data information: In (F), data are presented as means ± SEM. **P-value < 0.01 (ANOVA test). Data shown are representative results of two (A) or three (B, D, E, F) independent experiments.

Source data are available online for this figure.

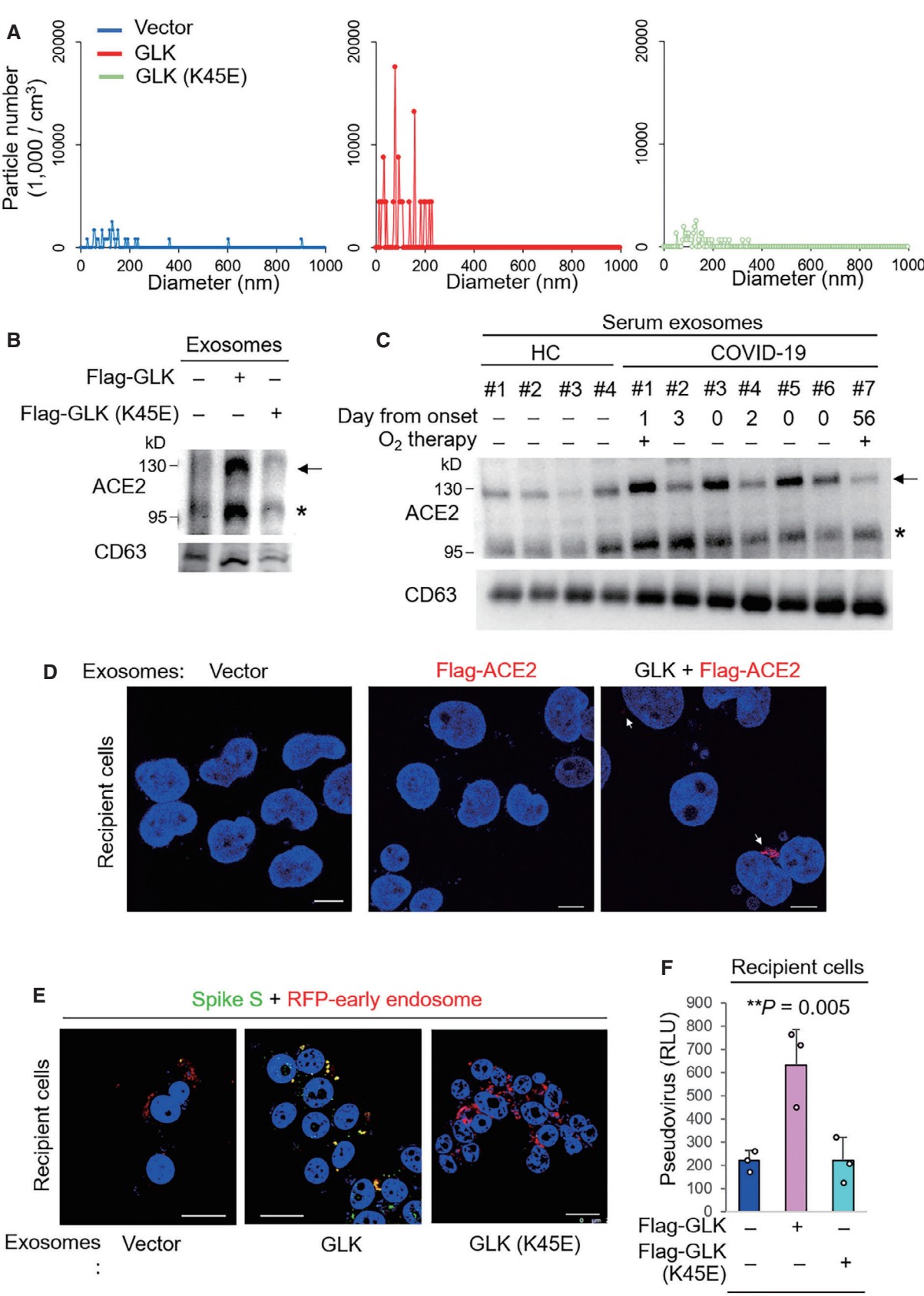

Figure 3.

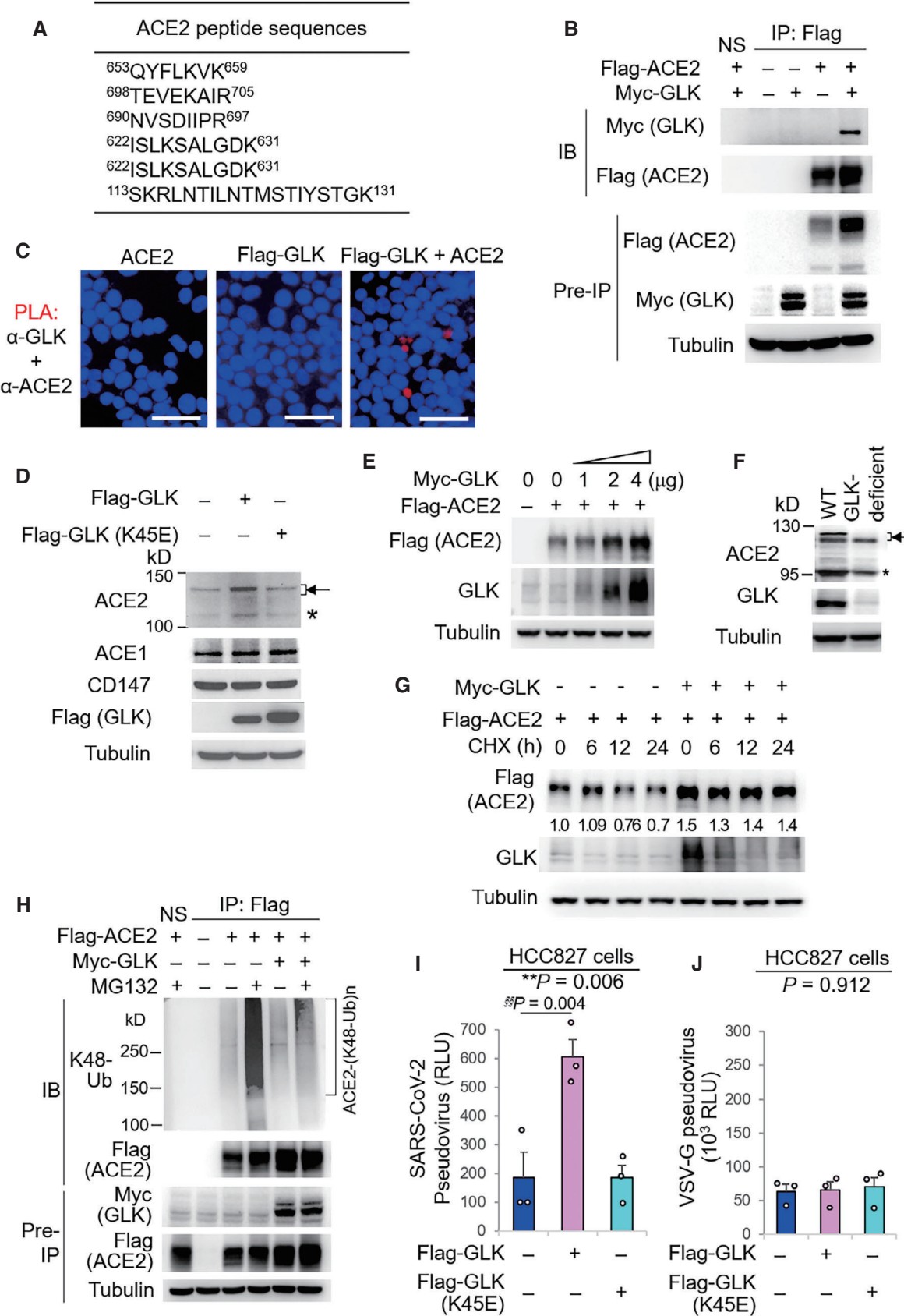

**Figure 4.**

◄

**Figure 4.  MAP4K3/GLK interacts with and stabilizes ACE2 proteins.**

A    The identified peptide sequences of the endogenous ACE2 proteins by mass spectrometry analyses using GLK immunocomplexes.

B    Co-immunoprecipitation of Flag-tagged ACE2 with Myc-tagged GLK proteins from lysates of HEK293T transfectants. NS, normal serum.

C    *In situ* PLA assays of the interaction between ACE2 and Flag-tagged GLK proteins in HEK293T cells. Nuclei were stained with DAPI. Red dots represent direct interaction signals. Original magnification, ×200. Scale bars, 50 μm.

D    Immunoblotting analyses of the endogenous ACE2, ACE1, CD147, and tubulin proteins, and the transfected Flag-GLK proteins in HCC827 lung epithelial cells, which were transfected with either GLK wild-type or kinase-dead (K45E) mutant. Arrowhead denotes glycosylated ACE2 proteins; asterisk denotes nonglycosylated ACE2 proteins.

E    Immunoblotting of Flag-tagged ACE2 (anti-Flag), Myc-tagged GLK (anti-GLK), and tubulin proteins from HEK293T cells co-transfected with Flag-ACE2 plus increasing amounts of Myc-GLK plasmids.

F    Immunoblotting of ACE2, GLK, and tubulin proteins in the lung tissues of wild-type or GLK-deficient mice. Arrowhead denotes glycosylated ACE2 proteins; asterisk denotes nonglycosylated ACE2 proteins.

G    Cycloheximide pulse-chase experiments in HEK293T cells. Immunoblotting of Flag-tagged ACE2 (anti-Flag), Myc-tagged GLK (anti-GLK), and tubulin proteins from HEK293T cells co-transfected with Flag-ACE2 and Myc-GLK. Transfected cells were treated with 100 μg/ml cycloheximide (CHX) for up to 24 h.

H    Flag-tagged ACE2 proteins were immunoprecipitated from lysates of HEK293T cells co-transfected with Flag-ACE2 and Myc-GLK, followed by immunoblotting with anti-Lys48-linked ubiquitination or anti-Flag antibody. Cells were treated with 25 μM MG132 for 2 h before being harvested. NS, normal serum.

I, J   Cell entry efficiencies of SARS-CoV-2 pseudovirus (I) or VSV-G pseudovirus (J) into GLK wild-type- or GLK (K45E)-overexpressing HCC827 epithelial cells were measured by luciferase activity and presented as relative light units (RLU) at 24 h postinfection. *n* = 3 (biological replicates).

Data information: In (I, J), data are presented as means ± SEM. **$P$-value < 0.01 (ANOVA test); §§$P$-value < 0.01 (Dunnett's test). Data shown are representative results of three (B–J) independent experiments.

Source data are available online for this figure.

release of exosomes, leading to the transport of functional ACE2 to other epithelial cells and thus enhancement of susceptibility to SARS-CoV-2 infection.

## GLK interacts with and stabilizes ACE2 proteins by attenuating ACE2 ubiquitination

To further investigate the role of GLK in ACE2 regulation, we characterized GLK-interacting proteins by mass spectrometry-based proteomics analyses. Surprisingly, proteomics analysis of GLK immunocomplexes revealed that ACE2 was a GLK-interacting protein (Fig 4A). Co-immunoprecipitation analyses confirmed the interaction between GLK and ACE2 in HEK293T cells (Fig 4B). *In situ* proximity ligation assays (PLA) with a combination of paired PLA probes corresponding to ACE2 and Flag showed PLA signals in cells co-transfected with ACE2 and Flag-GLK, suggesting a direct interaction (< 40 nm) between ACE2 and GLK (Fig 4C). Interestingly, GLK overexpression enhanced the endogenous ACE2, but not ACE1 nor CD147, protein levels in HCC827 lung epithelial cells (Fig 4D). As a control, overexpression of GLK (K45E) kinase-dead mutant did not increase ACE2 protein levels (Fig 4D), suggesting that the induction of ACE2 protein levels by GLK is dependent on GLK kinase activity. GLK enhanced Flag-tagged ACE2 protein levels in a dose-dependent manner in HEK293T cells (Fig 4E). Conversely, ACE2 protein levels were decreased in H661 lung epithelial cells by GLK shRNA knockdown (shRNA#2) or the GLK inhibitor (verteporfin) treatment (Appendix Fig S8A and B). Mouse ACE2 protein levels were decreased in the lungs of GLK-deficient mice compared to those of wild-type mice (Fig 4F). The two protein bands of the glycosylated mouse ACE2 proteins (about 130 kDa) may be due to proteolytic processing by proteases, such as TMPRSS2 (Heurich *et al*, 2014). Notably, Q–PCR data showed that GLK overexpression did not induce ACE2 mRNA levels (Appendix Fig S8C), suggesting a posttranslational regulation of ACE2 protein stability by GLK. To study whether GLK enhances ACE2 protein stability, the protein half-life of ACE2 was determined by cycloheximide pulse-chase experiments. GLK overexpression prolonged ACE2 protein half-life (estimated half-life: 39.1–157.5 h) in HEK293T cells (Fig 4G). We tested

whether GLK inhibits ubiquitin-mediated degradation of ACE2 protein. Immunoprecipitation experiments showed that Lys48-linked ubiquitination of ACE2 was significantly reduced by GLK overexpression (Fig 4H). We further studied whether GLK-induced ACE2 stabilization enhances SARS-CoV-2 infection. GLK-overexpressing epithelial cells were more efficiently infected with SARS-CoV-2 pseudovirus than that of GLK (K45E) kinase-dead mutant-overexpressing cells (Fig 4I). As a negative control, GLK overexpression did not enhance infection of VSV-G pseudovirus (Fig 4J). The data suggest that GLK increases ACE2 protein levels in epithelial cells, resulting in the enhancement of SARS-CoV-2 infection.

## GLK phosphorylates ACE2 at Ser776 and Ser783 residues

To investigate the mechanism of ACE2 stabilization by GLK, we first examined whether GLK induces ACE2 phosphorylation using Phos-tag immunoblotting. The data showed that GLK overexpression induced ACE2 phosphorylation, whereas GLK (K45E) kinase-dead mutant did not (Fig 5A). Next, we performed *in vitro* kinase assays to study whether GLK directly phosphorylates ACE2 using purified proteins. Serine phosphorylation signals of ACE2 proteins were enhanced by GLK but not GLK (K45E) kinase-dead mutant (Fig 5B). *In vitro* kinase assays and mass spectrometry analyses further showed that GLK-induced ACE2 phosphorylation sites at Ser776 and Ser783 residues (Fig 5C) but not at the previously reported AMPK-phosphorylated Ser680 residue (Zhang *et al*, 2018). Remarkably, GLK-induced ACE2 Ser776 and Ser783 phosphorylation events were also detected using mass spectrometry data of the serum exosomes from human COVID-19 patients in Cohort #3 (Fig 5D). Moreover, ACE2 Ser776 and Ser783 phosphorylation was also detected in the serum samples of COVID-19 patients from Cohort #4 (Appendix Fig S7B). The GLK-induced serine phosphorylation of ACE2 was significantly reduced by ACE2 (S776/783A) double mutations (Fig 5E), while the phosphorylation was modestly reduced by ACE2 (S776A) or (A783A) mutation alone (Fig 5E). Consistently, GLK-enhanced ACE2 protein levels were attenuated by ACE2 (S776/783A) mutation (Fig 5F). Conversely, ACE2 phosphomimetic (S776/783E, S776E, and S783E) mutants showed increased protein levels than wild-type

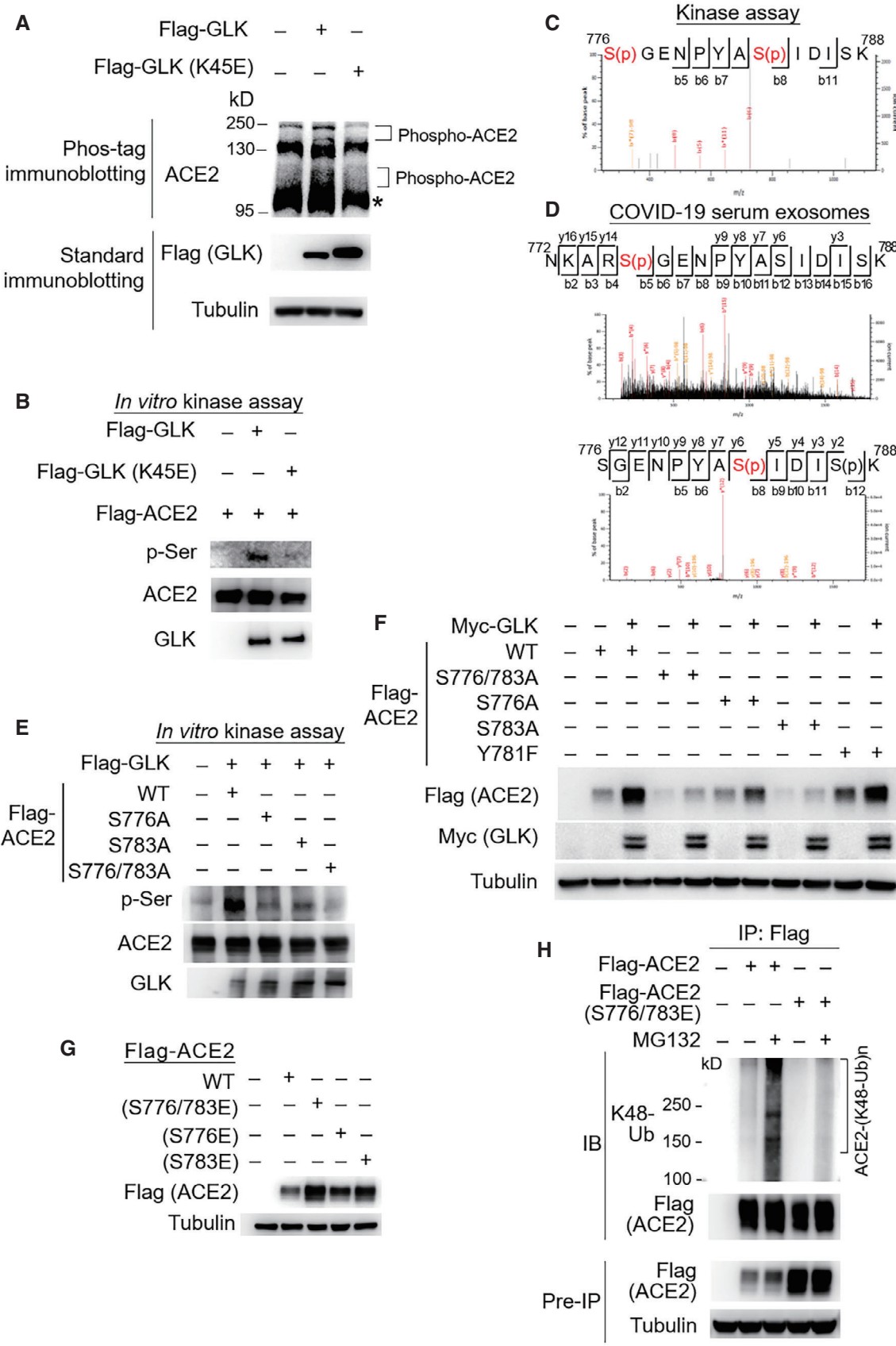

**Figure 5.**

◀

**Figure 5. MAP4K3/GLK phosphorylates and stabilizes ACE2 proteins.**

A   Immunoblotting analyses of ACE2 and Flag-tagged GLK in HCC827 cells transfected with either Flag-GLK or Flag-GLK (K45E) kinase-dead mutant. For immunoblotting of phosphorylated ACE2, Phos-tagged SDS–PAGE gel (Phos-tag™) was used, followed by immunoblotting with anti-ACE2 antibody. Asterisk denotes unglycosylated ACE2 proteins.

B   *In vitro* kinase assays using purified proteins. ACE2 serine phosphorylation, Flag-tagged GLK (anti-GLK), Flag-tagged GLK (K45E) kinase-dead mutant (anti-GLK), and Flag-tagged ACE2 proteins (anti-ACE2) were detected by immunoblotting.

C   Mass spectrometry analysis of GLK-phosphorylated ACE2 proteins after *in vitro* kinase assays. ACE2 Ser776 and Ser783 residues were phosphorylated by GLK wild-type but not GLK (K45E) kinase-dead mutant.

D   Mass spectrometry analysis of the ACE2 peptides from the serum exosomes of COVID-19 patients. The ACE2 protein peptide sequences containing phospho-Ser776 or phospho-Ser783 residue of ACE2 proteins detected in the serum exosomes of COVID-19 patients (Cohort #3) are shown. Exosomes were isolated using ExoQuick kits, and soluble proteins were removed by ExoQuick ULTRA columns. Exosomes were further purified by immunoprecipitation using a combination of anti-CD9, anti-CD63, and CD81 magnetic beads.

E   *In vitro* kinase assays using immunoprecipitated Flag-tagged ACE2 or Flag-tagged GLK immunocomplexes. ACE2 serine phosphorylation, ACE2, and GLK were detected by immunoblotting using anti-phospho-serine, anti-ACE2, and anti-GLK antibodies, respectively.

F   Immunoblotting analyses of Myc-tagged GLK, Flag-tagged ACE2, and tubulin proteins in HEK293T cells co-transfected with Flag-ACE2 wild-type or individual ACE2 (S776/783A, S776A, or S783A) mutants plus either empty vector or Myc-GLK.

G   Immunoblotting analyses of Flag-tagged ACE2 and tubulin proteins in HEK293T cells transfected with Flag-ACE2 wild-type or individual phosphomimetic ACE2 (S776/783E, S776E, or S783E) mutants.

H   Flag-tagged ACE2 proteins were immunoprecipitated from lysates of HEK293T cells transfected with Flag-ACE2 wild-type or a phosphomimetic ACE2 (S776/783E) mutant, followed by immunoblotting with anti-Lys48-linked ubiquitination or anti-Flag antibody. Cells were treated with 25 μM MG132 for 2 h before being harvested.

Data information: Data shown are representative results of three (A, B, E, F, G, H) independent experiments.
Source data are available online for this figure.

ACE2 (Fig 5G). As a negative control for the serine/threonine kinase GLK, ACE2 Tyr781 phosphorylation was not detected in ACE2 proteins from GLK *in vitro* kinase assay (Fig 5C), albeit it was detected in the ACE2 peptide from the serum samples of COVID-19 patients in Cohort #4 (Appendix Fig S7B). As expected, ACE2 (Y781F) mutation did not affect ACE2 protein levels (Fig 5F). Furthermore, Lys48-linked ubiquitination of ACE2 phosphomimetic (S776/783E) mutant was reduced compared to that of wild-type ACE2 (Fig 5H). These results suggest that GLK directly interacts with and phosphorylates ACE2, leading to ACE2 protein stabilization by attenuating ubiquitin-mediated protein degradation.

**GLK blocks the interaction of ACE2 with the E3 ligase UBR4**

To identify the E3 ligase that induces ACE2 Lys48-linked ubiquitination, Flag-ACE2 immunocomplexes were precipitated from transfected HEK293T cells and were subjected to mass spectrometry analyses. The E3 ligase UBR4 (ubiquitin N-recognin domain-containing E3 ligase 4) was identified as an ACE2-interacting protein with a high protein score (Figs 6A and EV2A). The protein score of UBR4 was drastically increased by the proteasome inhibitor MG132 treatment, whereas the UBR4 protein score was decreased by GLK overexpression (Fig 6A). The data suggest that UBR4 may be an E3 ligase that induces ACE2 ubiquitination, and the interaction between UBR4 and ACE2 may be blocked by GLK overexpression. Indeed, UBR4 overexpression induced ACE2 degradation (Fig 6B), which was reversed by MG132 treatment (Fig 6C). Moreover, UBR4 overexpression enhanced Lys48-linked ubiquitination of ACE2 (Fig 6D). These results suggest that UBR4 induces ACE2 Lys48-linked ubiquitination, leading to proteasomal degradation of ACE2. Co-immunoprecipitation data also showed an interaction between Flag-tagged ACE2 and Myc-tagged UBR4 (Fig 6D). The ACE2-UBR4 interaction was confirmed by *in situ* proximity ligation assays (PLA; Figs 6E and EV3A); the interaction was blocked by overexpression of wild-type GLK but not GLK kinase-

dead (K45E) mutant (Fig EV3A). Moreover, the ACE2-UBR4 interaction was further enhanced by ACE2 phospho-deficient (S776/783A) mutation but was abolished by ACE2 phosphomimetic (S776/783E) mutation (Fig 6E). Furthermore, *in vitro* ubiquitination of ACE2 by UBR4 was also enhanced by ACE2 phospho-deficient (S776/783A) mutation but attenuated by ACE2 phosphomimetic (S776/783E) mutation (Fig EV3B). The data suggest that GLK-induced ACE2 phosphorylation results in the dissociation between ACE2 and UBR4, leading to a reduction in ACE2 ubiquitination and degradation.

After re-analyzing ACE2 mass spectrometry data (Fig 6A) for putative ACE2 post-translational modification sites, Lys26, Lys94, Lys112, and Lys114 residues were identified as ACE2 ubiquitination sites (Figs 6F and EV2B). The ubiquitination of these four lysine residues was blocked by GLK overexpression (Fig 6A). To study whether ubiquitination of these lysine residues mediates UBR4-induced ACE2 degradation, the lysine residues of ACE2 were individually mutated to arginine residues. ACE2 (K26R), ACE2 (K112R), and ACE2 (K114R) mutants, but not ACE2 (K94R) mutant, were resistant to UBR4-induced ACE2 degradation (Fig 6G). Moreover, UBR4-induced ACE2 Lys48-linked ubiquitination was attenuated by a triple mutation (K26/112/114R) of ACE2 (Fig 6H). These results suggest that Lys26, Lys112, and Lys114 residues of ACE2 are responsible for UBR4-induced Lys48-linked ubiquitination and protein degradation of ACE2.

**SARS-CoV-2 induces a positive feedback loop of infection in humanized ACE2 mice**

To validate that GLK stabilizes ACE2 proteins and induces ACE2-containing exosomes to facilitate SARS-CoV-2 infection, we generated human ACE2 knockin mice (Fig 7A) and performed SARS-CoV-2 pseudovirus infection experiments. Human ACE2 knockin (hACE2 KI) was confirmed by PCR using mouse genomic DNAs and by Q–PCR using lung tissue mRNAs (Fig EV4A and B). Human ACE2

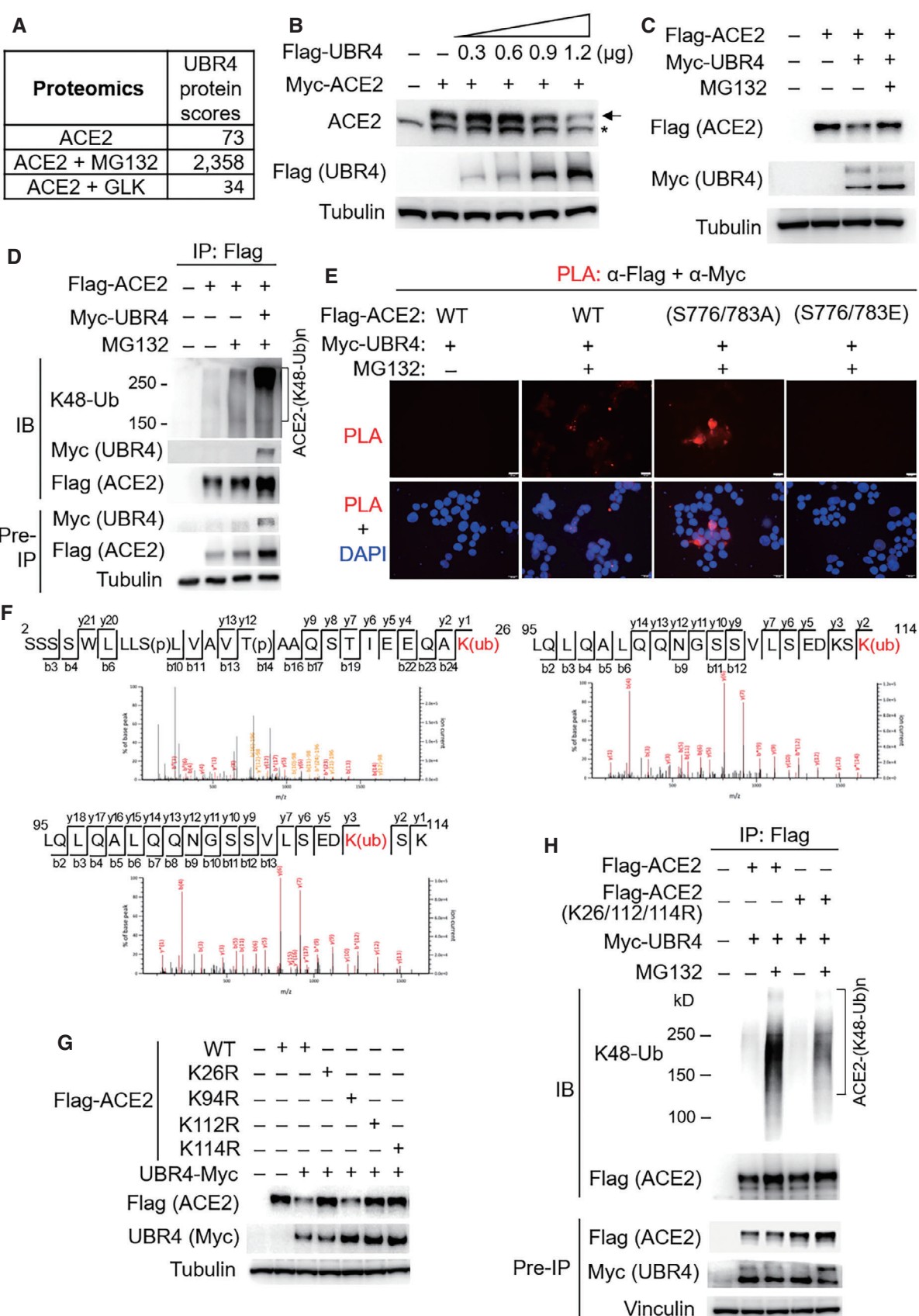

Figure 6.

**Figure 6. The E3 ligase UBR4 induces ACE2 ubiquitination and proteasomal degradation.**

A   The protein scores of UBR4 by mass spectrometry analyses using Flag-tagged ACE2 immunocomplexes from cells either treated with MG132 or co-transfected with Flag-ACE2 and Myc-GLK plasmids. The protein score is the sum of the highest ions score of MS/MS search for each distinct peptide.

B   Immunoblotting of Myc-tagged ACE2 (anti-ACE2), Flag-tagged UBR4 (anti-Flag), and tubulin proteins from HEK293T cells co-transfected with Myc-ACE2 and Flag-UBR4. Arrowhead denotes Myc-tagged ACE2 proteins; asterisk denotes endogenous ACE2 proteins.

C   Immunoblotting of Flag-tagged ACE2 (anti-Flag), Myc-tagged UBR4 (anti-Myc), and tubulin proteins from HEK293T cells co-transfected with Flag-ACE2 and Myc-UBR4. The co-transfected cells were treated with 25 μM MG132 for 2 h before being harvested.

D   Flag-tagged ACE2 proteins were immunoprecipitated from lysates of HEK293T cells co-transfected with Flag-ACE2 and Myc-UBR4, followed by immunoblotting with anti-Lys48-linked ubiquitination, anti-Myc, or anti-Flag antibody. Cells were treated with 25 μM MG132 for 2 h before being harvested.

E   *In situ* PLA assays of the interaction between Flag-tagged ACE2 (wild-type or mutant) and Myc-tagged UBR4 proteins in HEK293T cells treated with 25 μM MG132 for 2 h. Nuclei were stained with DAPI. Red dots represent direct interaction signals. Original magnification, ×200. Scale bars, 20 μm.

F   Mass spectrometry analysis of the ACE2 peptides containing ubiquitination residues from Flag-ACE2-transfected HEK293T cells treated with 25 μM MG132 for 2 h.

G   Immunoblotting analyses of Flag-tagged ACE2, Myc-tagged UBR4, and tubulin proteins in HEK293T cells co-transfected with Flag-ACE2 wild-type or individual (K26R, K94R, K112R, or K114R) mutants plus either empty vector or Myc-UBR4.

H   Flag-tagged ACE2 proteins were immunoprecipitated from lysates of HEK293T cells co-transfected with Flag-ACE2 wild-type or Flag-ACE2 (K26/112/114R) mutant plus Myc-UBR4 plasmids, followed by immunoblotting with anti-Lys48-linked ubiquitination or anti-Flag antibody. Cells were treated with 25 μM MG132 for 2 h before being harvested.

Data information: Data shown are representative results of three (A, B, C, D, E, G, H) independent experiments.
Source data are available online for this figure.

mRNA levels were detected in multiple organs of hACE2 KI mice (Fig EV4B). Consistent with reported data (Wiener *et al*, 2007; Salamanna *et al*, 2020), human ACE2 proteins were highly expressed in the lungs, kidneys, spleen, and lymph nodes of hACE2 KI mice (Fig EV4C). Human ACE2 knockin mice were successfully infected with SARS-CoV-2 pseudovirus (Fig 7B), whereas wild-type mice were not infected due to the lack of human ACE2 proteins (Fig 7B). We found that GLK protein levels in the lung tissues of hACE2 KI mice were increased by SARS-CoV-2 pseudovirus infection (Fig 7C), while ACE2 levels were also induced in the lung tissues (Fig 7C). The spike proteins were detected in the lung tissues of the infected hACE2 KI mice (Fig 7D), while GLK levels were concomitantly increased in the infected lung tissues (Fig 7D). Moreover, ACE2 protein levels in the lung tissue, and in the exosomes isolated from the sera and bronchoalveolar lavage fluid (BALF) were increased in the

infected hACE2 KI mice compared to those of wild-type mice at 4 days post pseudovirus infection (Figs 7E and EV4D). To confirm that the induced ACE2 proteins are indeed exosomal proteins but not soluble ACE2 proteins, we performed the *in situ* proximity ligation assay (PLA) using anti-human ACE2 antibody plus anti-CD9 antibody to detect close proximity (< 40 nm) between hACE2 and the exosome marker CD9. The PLA signals representing ACE2-containing exosomes were detected in the lung tissues of hACE2 KI mice infected with SARS-CoV-2 pseudovirus (Fig 7F).

To further validate the results derived from pseudovirus infection studies, we performed live SARS-CoV-2 infection using human ACE2 transgenic (EF1α-hACE2 Tg) mice (Fig 7G). After infection of live SARS-CoV-2, significant amounts of pulmonary viremia were observed in infected EF1α-hACE2 Tg mice but not in infected wild-type mice (Fig EV4E). Live SARS-CoV-2-infected EF1α-hACE2 Tg

**Figure 7. SARS-CoV-2 induces a positive feedback loop of infection in mice.**

A   Schematic diagram of the human ACE2 knockin (hACE2 KI) mutant allele. The knockin with hACE2 cDNA plus SV40 late poly A signal results in blocking the transcription of mouse ACE2 (mACE2). The box with numbers, the exons of mACE2; arrowheads, the primers for genotyping PCR.

B–F   Wild-type and hACE2 KI mice were intranasally infected with SARS-CoV-2 pseudovirus. Infection efficiencies of SARS-CoV-2 pseudovirus into mouse tissues were measured by IVIS and presented as luminescence counts at 4 days postinfection (B). Immunoblotting of ACE2, GLK, and tubulin proteins in the lung tissues of infected or noninfected mice (C). Representative immunohistochemistry of the SARS-CoV-2 spike protein (in red) and GLK protein (in green) in the lung tissues of the infected hACE2 KI and wild-type mice (D). Scale bars, 20 μm (D). Immunoblotting of CD9 and mouse/human ACE2 in the exosomes isolated from the BALFs (E) of the infected mice. Exosomes were isolated sequentially using ExoQuick kits and then ExoQuick ULTRA columns. The hACE2-containing exosomes in the lung tissues of wild-type and hACE2 KI mice were determined by *in situ* proximity ligation assays (PLA) of close proximity (< 40 nm) between hACE2 and the exosome marker CD9 using anti-human ACE2 antibody plus anti-CD9 antibody (F). Scale bars, 20 μm (F).

G   Schematic depiction of the construction of the EF-1α promoter-driven, murine codon-optimized human ACE2 (hACE2) cDNA with bovine growth hormone (bGH) poly A signal.

H–J   Human ACE2 transgenic (EF1α-hACE2 Tg) mice were intranasally infected with 2 × 10^5 pfu of live SARS-CoV-2. The survival rates in EF1α-hACE2 Tg mice challenged with/without SARS-CoV-2 were monitored (H). The lung tissues of infected miceon day 3 postinfection were collected and then analyzed by immunohistochemistry (IHC) and PLA assays. Representative IHC data of hACE2 (in red) and GLK (in green) in the lung tissues of the infected wild-type and EF1α-hACE2 Tg mice were shown (I). The hACE2-containing exosomes in the lung tissues of infected wild-type and EF1α-hACE2 Tg mice were determined by PLA (J). Cell nuclei were stained with DAPI. Original magnification, ×630. Scale bars, 10 {m.

K, L   Adoptive transfer of ACE2-containing exosomes facilitates SARS-CoV-2 pseudovirus infection. Serum exosomes (exo) isolated from wild-type (WT) mice and hACE2 KI/activated GLK transgenic (PolII-GLK E351K Tg) mice were adoptively transferred into wild-type recipient mice every 3 days for 12 days, followed by SARS-CoV-2 pseudovirus infection. Infection efficiencies in mouse tissues were measured by IVIS and presented as luminescence counts at 3 days postinfection (K). hACE2-containing exosomes in the lung tissues of exosome-recipient wild-type mice were determined by PLA using anti-human ACE2 antibody plus anti-CD9 antibody (L). Scale bars, 10 {m (L).

Data information: Arrowhead denotes glycosylated ACE2 proteins; asterisk denotes nonglycosylated ACE2 proteins. WT, wild-type mice; KI, hACE2 knockin mice. Tg, transgenic mice.
Source data are available online for this figure.

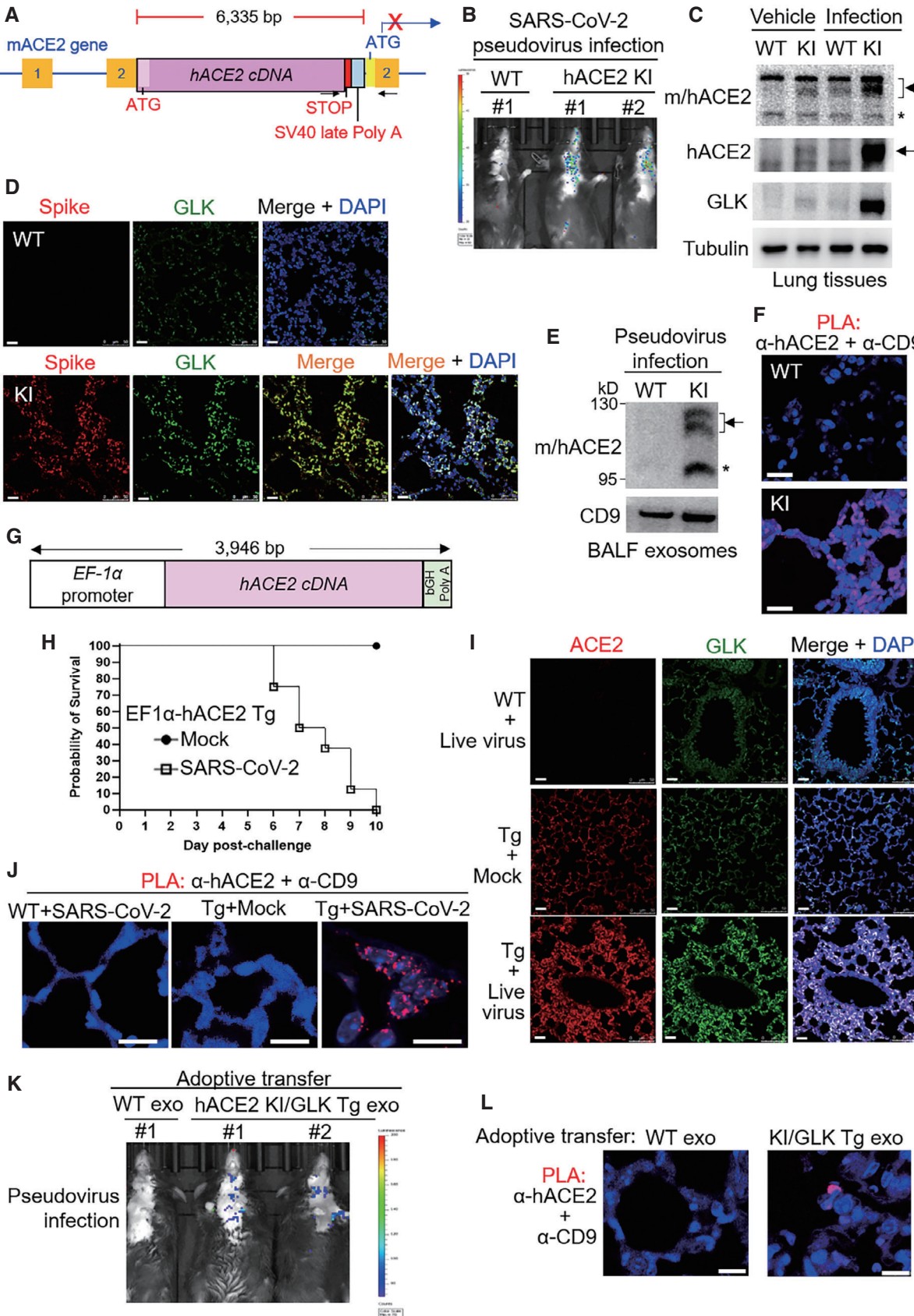

**Figure 7.**

mice displayed severe weight loss (Fig EV4F) and all died on day 10 postinfection (Fig 7H), whereas mock-infected EF1α-hACE2 Tg mice displayed normal body weight gain (Fig EV4F) and 100% survival rate (Fig 7H). Furthermore, immunohistochemistry (IHC) staining showed that ACE2 and GLK levels were enhanced in live SARS-CoV-2-infected EF1α-hACE2 Tg mice compared to those of infected wild-type mice (Fig 7I). PLA data showed that hACE2-containing exosomes were also detected in the lung tissues of live SARS-CoV-2-infected EF1α-hACE2 Tg mice (Fig 7J).

### Adoptive transfer of ACE2-containing exosomes facilitates SARS-CoV-2 pseudovirus infection

To demonstrate that hACE2-containing exosomes facilitate virus infection, we performed adoptive transfer using serum exosomes from hACE2 knockin/GLK transgenic mice, followed by pseudovirus infection. After the adoptive transfer of hACE2-containing exosomes, wild-type recipient mice became susceptible to SARS-CoV-2 pseudovirus infection (Fig 7K). Furthermore, PLA data showed the presence of hACE2-containing exosomes from hACE2 KI/GLK Tg mice, but not from wild-type mice, in the lung tissue of wild-type recipient mice (Fig 7L). Collectively, these results suggest that spike proteins of SARS-CoV-2 may stimulate GLK-induced ACE2 stabilization and ACE2-containing exosome production in the infected lung tissues, hence exacerbating SARS-CoV-2 infection.

### GLK inhibitor attenuates ACE2 protein levels, ACE2-containing exosomes, and SARS-CoV-2 pseudovirus infection

To study whether inhibition of GLK attenuates SARS-CoV-2 infection, the infected hACE2 KI mice were treated with a GLK inhibitor (verteporfin; Chuang et al, 2019b). Remarkably, SARS-CoV-2 pseudovirus infection in hACE2 KI mice was blocked by the GLK inhibitor (verteporfin) treatment (Fig EV5A). The ACE2 protein levels in the lung tissues of pseudovirus-infected hACE2-KI mice were also decreased by the GLK inhibitor (verteporfin) treatment (Fig EV5B). Moreover, ACE2-containing exosomes were reduced in the serum and BALF of the infected hACE2-KI mice (Fig EV5C and D). These results support the notion that the SARS-CoV-2 spike protein induces GLK levels in epithelial cells and subsequent induction of ACE2 protein stability and ACE2-containing exosomes, which may enhance SARS-CoV-2 infection of other epithelial cells.

Taken together, these results support that SARS-CoV-2 infection induces GLK overexpression and subsequent ACE2 stabilization, and ACE2-containing exosome production, resulting in a positive feedback loop of SARS-CoV-2 infection.

## Discussion

In summary, GLK overexpression in epithelial cells is correlated with COVID-19 severity and may play an important role in COVID-19 pathogenesis. SARS-CoV-2 S protein increases GLK levels in epithelial cells, in pseudovirus-infected hACE2 KI mice, and in live SARS-CoV-2-infected EF1α-hACE2 Tg mice. GLK phosphorylates and stabilizes ACE2 protein in lung epithelial cells by reducing UBR4-mediated ubiquitination and proteasomal degradation. Moreover, GLK-overexpressing epithelial cells shed ACE2-containing exosomes

to other epithelial cells, resulting in increased ACE2 proteins and enhanced SARS-CoV-2 infection of otherwise less susceptible cells.

Simultaneous co-treatment of ACE2-containing extracellular vesicles from HEK293FT cells with SARS-CoV-2 pseudovirus was reported to have a decoy effect in attenuating infection (Cocozza et al, 2020). This decoy effect is likely due to the competitive binding for SARS-CoV-2 spike proteins by surface ACE2 proteins on exosomes in vitro. In contrast, in our sequential treatment studies, ACE2-containing exosomes from GLK-overexpressing cells facilitated SARS-CoV-2 pseudovirus infection after the transport of exosomal ACE2 proteins into recipient cells. Furthermore, adoptive transfer of hACE2-containing exosomes allows wild-type mice to become susceptible to SARS-CoV-2 pseudovirus infection. Consistently, ACE2-containing serum exosomes were increased in COVID-19 patients but not in healthy controls; thus, our data are further supported by data of serum exosomes from COVID-19 patients. We cannot rule out the possibility that SARS-CoV-2 infection may induce epithelial cell damage in COVID-19 patients, resulting in enhancement of exosome release. In addition, our findings may in part provide an explanation for why ACE2 mRNA levels are very low in airway epithelial cells using scRNA-seq (Sungnak et al, 2020), whereas SARS-CoV-2 infection of airway epithelium is quite efficient.

Our findings suggest that GLK overexpression in epithelial cells is a critical step of SARS-CoV-2 infection in patients. Multiple organs are infected with SARS-CoV-2 in COVID-19 patients (Liu et al, 2021). Furthermore, our in vitro and in vivo data suggest that upon SARS-CoV-2 infection, the spike protein induces GLK overexpression in epithelial cells, which may facilitate further infection of other cells. Conversely, the GLK inhibitor verteporfin attenuated ACE2 protein levels in vitro and in vivo; verteporfin also decreased ACE2-containing exosomes and blocked SARS-CoV-2 pseudovirus infection in hACE2 KI mice. These results suggest that verteporfin could be repositioned to treat COVID-19. Besides being the SARS-CoV-2 entry receptor, our findings also raise an interesting possibility that the spike protein may contribute to COVID-19 pathogenesis by inducing signaling molecules such as MAP4K3/GLK. In addition to COVID-19 vaccine adjuvants, the adverse effects may also be induced by the spike protein-stimulated MAP4K3/GLK or other signaling molecules. It would be interesting to explore the potential pathological roles of the spike protein in inflammation induced by SARS-CoV-2 virus.

Besides coronavirus infection, ACE2 overexpression is also correlated with several human diseases, including hypertension, diabetes, and cardiovascular diseases (Hamming et al, 2007; Narula et al, 2020). Thus, MAP4K3/GLK may be involved in the pathogenesis of coronavirus infection, hypertension, diabetes, or cardiovascular diseases.

## Materials and Methods

### Human samples

This study was conducted in accordance with the Helsinki Declaration. A total of 23 individuals, including 10 healthy individuals and 13 COVID-19 patients were enrolled in this study (Cohort #3). Serum samples were obtained from National Health Research

Institutes (NHRI) Biobank for COVID-19 patients in Taiwan (TLCRC #20-004); all study participants provided written informed consent. No Taiwanese indigenous peoples were enrolled in this cohort. Sample collection from healthy controls and COVID-19 patients, and experiments were approved by the ethics committee of the National Health Research Institutes (#EC1090509). The experiments conformed to the principles set out in the Department of Health and Human Services Belmont Report.

### Single-cell RNA-sequencing data analysis

Single-cell RNA-sequencing (scRNA-seq) data (gene read counts) from two independent cohorts (Chua *et al*, 2020; Liao *et al*, 2020) were used in this study. ScRNA-seq data of human BALF cells from three healthy controls and nine COVID-19 patients (Cohort #1; Liao *et al*, 2020) were downloaded from Gene Expression Omnibus (GEO; accession number GSE145926). ScRNA-seq data of nasopharyngeal cells from five healthy controls and 19 COVID-19 patients (Cohort #2; Chua *et al*, 2020) were downloaded from FigShare: https://doi.org/10.6084/m9.figshare.12436517. The data were analyzed by the software BD Rhapsody (BD Biosciences) plus the R package Seurat. Cells that had less than 200 genes were excluded. Dimensionality reduction was performed using Uniform Manifold Approximation and Projection (UMAP); clustering analysis was performed according to individual subsets of variable genes.

### Mice

All animal experiments were performed in the AAALAC-accredited animal housing facilities at National Health Research Institutes (NHRI). All mice were used according to the protocols and guidelines approved by the Institutional Animal Care and Use Committee of NHRI (#IACUC109123 and #IACUC109135). GLK-deficient mice and activated GLK transgenic (PolII-GLK E351K Tg) mice in C57BL/6J background were generated as described previously (Chuang *et al*, 2011, 2019a). All mice used in this study were maintained in temperature-controlled and pathogen-free cages.

### Generation of human ACE2 knockin mouse line

Human ACE2 knockin mouse line was generated using 2-cell homologous recombination CRISPR approach as described previously (Gu *et al*, 2020). In brief, the 6,335-bp knockin insert containing human ACE2 cDNA and SV40 late Poly A (Orozco *et al*, 2002) was constructed by gene synthesis (Genewiz) using optimized mouse codons, followed by PCR amplification with 5′-biotin modification. Cas9 mRNA was synthesized by using pCS2$^+$-Cas9-mSA plasmid (Addgene #103882) as a template for *in vitro* transcription with the mMESSAGE mMACHINE SP6 transcription kit (Ambion). The sgRNA was designed by the CRISPR design tool on Benchling website and produced by chemical synthesis (Synthego). Cas9-mSA mRNA (100 ng/μl), synthetic sgRNA (5 ng/μl), and insert DNA (20 ng/μl) were mixed in microinjection buffer (10 mM Tris–HCl, pH 7.4, and 0.25 mM EDTA) and then microinjected into C57BL/6J mouse 2-cell stage embryos collected at 44 h after human chorionic gonadotropin (hCG) injection. After microinjection, 25–30 injected embryos were transferred into the oviducts of embryonic day 0.5 (E0.5) pseudo-pregnant females in CD-1 background. Seven days

postbirth, the pups were genotyped by long-range PCR with the primers targeting the insert and mouse ACE2 exon 2. hACE2 KI mice were used for SARS-CoV-2 pseudovirus infection experiments. hACE2 KI mice were crossed with activated GLK transgenic (PolII-GLK E351K Tg) mice, and the offspring (hACE2 KI/GLK Tg) mice were used for exosome adoptive transfer experiments.

### Generation of human ACE2 transgenic (EF1α-hACE2 Tg) mouse line

The cytomegalovirus (CMV) promoter region in pcDNA3.1(−) plasmid was replaced by the human elongation factor 1α (EF-1α) promoter to generate the pcDNA3.1(−)/EF1α plasmid. The murine codon-optimized human ACE2 (hACE2) cDNA (Table EV1) was inserted into the multiple cloning sites of pcDNA3.1(−)/EF1α plasmid to generate pcDNA3.1-hACE2. Linearized pcDNA3.1-hACE2 DNAs were microinjected into C57BL/6J mouse pronucleus zygotes and then were transplanted into pseudo-pregnant female ICR mice. The genomic DNAs were subjected to PCR by KAPA Taq ReadyMix PCR Kit (Kapa Biosystems) with a pair of hACE2 primers; forward primer: GCC ACT GTA CGA GGA GTA C, reverse primer: CCT CTG CTG TAA TCG TAT CCG (Table EV1). The EF1α-hACE2 Tg mouse line ubiquitously expressing hACE2 was used for live SARS-CoV-2 infection experiments.

### SARS-CoV-2 pseudovirus infection mouse model

Wild-type and hACE2 KI mice were anesthetized with isoflurane for 3 min and intranasally infected with 30 μl SARS-CoV-2 pseudovirus ($1 \times 10^7$/ml; MBS434275, MyBioSource) on Day 0 and Day 1. To detect pseudovirus-infected tissues, mice were injected with luciferin (Sigma-Aldrich) and subjected to a noninvasive *in vivo* imaging system (IVIS) at 4 days postinfection. For treatment of the GLK inhibitor, mice were treated with the GLK inhibitor verteporfin (10 {M in 100 {l PBS; Sigma-Aldrich) by intraperitoneal injection on Day 0.

### Live SARS-CoV-2 infection mouse model

Wild-type and EF1α-hACE2 transgenic mice (8–12 weeks) were maintained in pathogen-free cages. The live SARS-CoV-2 challenge studies were performed in the P3 laboratory of NHRI (#IBC109133). Mice were administered intranasally with $2 \times 10^5$ Median Tissue Culture Infectious Dose (TCID$_{50}$) of SARS-CoV-2 (original strain), and then body weights and survival rates were monitored daily for 10 days. To examine the pulmonary virus loads, mice were sacrificed on days 3 and 6 after the viral challenge, and the lung tissues were collected/homogenized for detection of virus titers using TCID$_{50}$ assay on Vero cells. Briefly, Vero cells were incubated with a serial dilution of lung homogenates and then cultured in M119 medium with 5% FBS for 5 days. Using fold dilution and optical microscopy, 50% of lysed cells were judged and calculated to obtain the virus titer in the lung tissues.

### Cell lines and transfection

The human lung cancer cell lines (HCC827 and H661; ATCC) were cultured in RPMI-1640 medium; the human embryonic kidney cell

line (HEK293T; ATCC) was cultured in DMEM medium. Both media were supplemented with 10% fetal bovine serum, 100 U/ml penicillin, and 100 mg/ml streptomycin (Gibco). All cells were free of mycoplasma contamination and grown at 37°C in a humidified atmosphere of 5% $CO_2$ in air. Plasmids were transfected into cells using polyethylenimine reagents (Sigma-Aldrich).

## Reagents, antibodies, and plasmids

Cycloheximide was purchased from Sigma-Aldrich. Anti-ACE2 (clone A4612, 1:1,000) antibody was purchased from ABclonal. Anti-ACE2 (clone OTI1D6, 1:1,000) antibody (Fig 3B and C) was purchased from Cell Signaling. Anti-human ACE2 (clone EPR4436, 1:1,000) antibody (Fig 7D and F) was purchased from Abcam. Anti-SARS-CoV-2 spike protein antibody (GTX635654, 1:1,000) was purchased from GeneTex. Anti-Myc (clone 9E10, 1:10,000), anti-Flag (clone M2, 1:4,000), and anti-tubulin (clone AA2, 1:5,000) antibodies were purchased from Sigma-Aldrich. Anti-phospho-serine (clone 4A4, 1:500) antibody was purchased from Millipore. Anti-Lys48-linked ubiquitination (clone Apu2, 1:1,000) antibody was purchased from Merck. Anti-GLK monoclonal antibody (clone C3, 1:6,000), and GLK or GLK (K45E) kinase-dead mutant plasmids were reported previously (Chuang et al, 2018). Anti-CD9 (clone 4H7B9, 1:1,000) antibody was purchased from Proteintech. The plasmid encoding 3xMyc-tagged human UBR4 cDNA was generated by cloning the cDNA from HCC827 cells by RT–PCR and subcloning into the vector pCMV-3Tag-9 (Agilent Technologies). The plasmids encoding 3xFlag-tagged or 3xMyc-tagged human ACE2 cDNA (NCBI accession number: NM_021804.3) were generated by cloning the cDNA from HCC827 cells by RT–PCR and then subcloning into the vector pCMV6-AC-3DDK (Origene) or pCMV-3Tag-9 (Agilent Technologies). The plasmids encoding ACE2 (S776A), ACE2 (S783A), ACE2 (S776/783A), ACE2 (S776E), ACE2 (S783E), ACE2 (S776/783E), and ACE2 (Y781F) were generated by mutating the indicated serine or tyrosine residue to alanine, glutamic acid, or phenylalanine residue on the 3xFlag-tagged ACE2 plasmid. The plasmids encoding ACE2 (K26R), ACE2 (K94R), ACE2 (K112R), ACE2 (K114R), and ACE2 (K26/112/114R) were generated by mutating the indicated lysine residue to arginine residue on the 3xFlag-tagged ACE2 plasmid. Cell-Light® Early Endosomes-RFP reagent was purchased from Thermo-Fisher. SARS-CoV-2 spike protein (MBS8574721), S1 protein (MBS553728), S2 protein (MBS2563874), luciferase-expressing SARS-CoV-2 pseudovirus (MBS434275), and negative-control pseudovirus (MBS434280) were purchased from MyBioSource. Phos-tag™ acrylamide was purchased from Fujifilm. Real-time PCR primers and probe for murine GLK, forward primer: GGT GTG GTC ATA TTA CAA, reverse primer: CAG GAT TTA TGT TCA TAT TAC, probe: TGC TGG CAA TGA ACC AAG ACA. Real-time PCR primers for the codon-optimized human ACE2, forward primer: GCA GCT CAA CCT TTC CTC CT, reverse primer: GTG GCA GCA GAC AGA CTC AT.

## Extracellular vesicle (exosome) isolation

Two-hundred microlitre exosome precipitation solution (ExoQuick kit; System Biosciences) were added to 1,000 μl cultured supernatants of GLK wild-type- or GLK (K45E) kinase-dead mutant-overexpressing HCC827 lung epithelial cells. For human serum

samples of COVID-19 patients from Cohort #3, 20 μl exosome precipitation solution was added to 100 μl sera. The ExoQuick and supernatant mixtures were co-incubated at 4°C overnight, followed by centrifugation at 3,000 g for 15 min. To remove soluble proteins, precipitants were resuspended and loaded onto ExoQuick ULTRA columns, followed by centrifugation at 1,000 g for 30 s. The isolated extracellular vesicles (exosomes) from cell lines were subjected to ZetaView nanoparticle tracking analysis (Fig 3A), immunoblotting (Fig 3B), or exosome transfer experiments (Fig 3D and F). For experiments using human serum exosomes of COVID-19 patients from Cohort #3 (Figs 3C and EV1), extracellular vesicles were further purified using anti-CD9 and anti-CD63 plus anti-CD81 magnetic beads (Exo-Flow96 Tetra IP kit, System Biosciences), followed by elusion using 20 μl exosome elusion buffer (System Biosciences).

## Liquid chromatography-mass spectrometry and data analysis

For identification of GLK-interacting proteins, immunocomplexes of Flag-tagged GLK were immunoprecipitated by anti-Flag antibody (M2; Sigma-Aldrich) from lysates of HEK293T cells transfected with vector or Flag-GLK plasmid. For the identification of GLK-induced ACE2 phosphorylation residues, immunocomplexes of Myc-tagged ACE2 were immunoprecipitated by anti-Myc antibody (9E10; Sigma-Aldrich) from lysates of HEK293T cells co-transfected with Myc-ACE2 plus either Flag-GLK or Flag-GLK (K45E) kinase-dead mutant plasmids. Higher (> 150 kDa) and lower (< 150 kDa) molecular-weight protein bands were collected from Instant blue (GeneMark)-stained SDS–PAGE gels. Proteins were digested with trypsin and subjected to LC–MS/MS analyses by LTQ-Orbitrap Elite hybrid mass spectrometer (for identifying GLK-interacting proteins) or LTQ-Orbitrap Fusion hybrid mass spectrometer (for mapping ACE2-phosphorylated residues) using approaches described previously (Chuang et al, 2018, 2019a). The peptide data were analyzed by MASCOT MS/MS Ions Search (Matrix Science) under the following condition: peptide mass tolerance, 20 ppm; fragment MS/MS tolerance, 0.6 Da; allow up to one missed cleavage; peptide charge, $2^+$, $3^+$, and $4^+$.

For characterization of exosomal proteins in the serum exosomes of COVID-19 patients from Cohort #3, exosomal proteins were digested with trypsin and subjected to LC–MS/MS analyses by LTQ-Orbitrap Fusion hybrid mass spectrometer. The peptide data were analyzed by MASCOT MS/MS Ions Search (Matrix Science) under the following condition: peptide mass tolerance, 20 ppm; fragment MS/MS tolerance, 1.2 Da; allow up to two missed cleavage; peptide charge, $2^+$, $3^+$, and $4^+$. Serum exosomal proteins identified from COVID-19 patients and healthy controls are listed in Dataset EV1.

For analyzing mass data of serum samples from 21 healthy controls, 24 mild COVID-19 patients, and 20 severe COVID-19 patients (Cohort #4; Shen et al, 2020), 320 mass spectrometry raw data (320 files in total) were downloaded from ProteomeXchange Consortium (https://www.iprox.org/; project ID IPX0002106000). Every 40 of 320 raw data were consolidated into eight individual experiment batches by Mascot Distiller. Peptide data of individual batches were analyzed by MASCOT MS/MS Ions Search (Matrix Science) under the following condition: quantitation, TMTpro 16plex; peptide mass tolerance, 40 ppm; fragment mass MS/MS tolerance, 1.2 Da; max missed cleavages, 2; peptide charge, $2^+$, $3^+$, and $4^+$.

**Immunoprecipitation, immunoblotting analysis, *in situ* proximity ligation assay (PLA), and *in vitro* kinase assay**

These experiments were performed as described previously (Chuang *et al*, 2018, 2019a).

**Statistics**

*In vivo* experiments were conducted using distinct samples; *in vitro* experiments were performed in at least three independent experiments. Statistical analyses were performed by using Excel, SPSS, or BD SEQGEQ. No randomization or blinding experiment was done. Three groups were compared by the ANOVA test, followed by the Dunnett's test. The Kruskal–Wallis test was used to analyze violin plots. A nonparametric Mann–Whitney test was used to analyze the body weight of hACE2 transgenic mice. *P*-values lower than 0.05 were considered to be significant.

# Data availability

This study includes no data deposited in external repositories. The data supporting the findings of this study are documented within the paper and are available from the corresponding author upon request.

**Expanded View** for this article is available online.

## Acknowledgements
We thank the Institute of Biological Chemistry of Academia Sinica for mass spectrometry using GLK or ACE2 immunocomplexes. We thank the Transgenic Mouse Core of NHRI for the generation of EF1α-hACE2 transgenic mice and hACE2 knockin mice. We thank the Core Instrument Center of the National Health Research Institutes (NHRI), Taiwan, for technical support in confocal cell imaging and consolidation of mass data using patient samples (Cohort #4). We thank NHRI Biobank for providing COVID-19 serum samples. We also thank Taiwan Bioinformatics Institute Core Facility at NHRI for technical support in file-format transfer of scRNA-seq data from Cohort #2. This work was supported by grants from the National Health Research Institutes, Taiwan (IM-107-SP-01 to T-HT), and the Ministry of Science and Technology, Taiwan (MOST-110-2320-B-400-018 to T-HT). T-HT is a Taiwan Bio-Development Foundation (TBF) Chair in Biotechnology.

## Author contributions
**Huai-Chia Chuang:** Conceptualization; data curation; software; formal analysis; supervision; validation; investigation; visualization; methodology; writing – original draft; project administration; writing – review and editing. **Chia-Hsin Hsueh:** Data curation; software; formal analysis; investigation; methodology. **Pu-Ming Hsu:** Data curation; formal analysis; investigation; methodology. **Rou-Huei Huang:** Data curation; software; formal analysis. **Ching-Yi Tsai:** Data curation; formal analysis; investigation; methodology. **Nai-Hsiang Chung:** Data curation; formal analysis. **Yen-Hung Chow:** Conceptualization; resources; data curation; formal analysis; supervision; investigation; methodology; writing – review and editing. **Tse-Hua Tan:** Conceptualization; resources; formal analysis; supervision; funding acquisition; validation; investigation; visualization; methodology; project administration; writing – review and editing.

**The paper explained**

**Problem**

SARS-CoV-2 infection in humans induces severe and acute respiratory disease. The mRNA levels of the SARS-CoV-2 entry receptor ACE2 are generally low in airway epithelial cells; however, SARS-CoV-2 infection of airway epithelium is quite efficient.

**Results**

Our study reveals the role of the kinase MAP4K3 (also named GLK) in enhancing SARS-CoV-2 infection susceptibility of epithelial cells. Upon SARS-CoV-2 infection, the spike protein of SARS-CoV-2 increases MAP4K3 (GLK) levels in epithelial cells. MAP4K3 (GLK) stabilizes ACE2 proteins and induces ACE2-containing exosome release, resulting in increased ACE2 proteins and enhanced SARS-CoV-2 infection of otherwise less susceptible cells. Collectively, ACE2 stabilization by SARS-CoV-2-induced MAP4K3 (GLK) may contribute to the pathogenesis of COVID-19.

**Impact**

Our study demonstrates that MAP4K3 (GLK) overexpression in epithelial cells is a critical step of SARS-CoV-2 infection in COVID-19 patients. The findings suggest that targeting MAP4K3 (GLK)-induced ACE2 proteins in epithelial cells could be a novel strategy to prevent or treat COVID-19.

**Disclosure and competing interests statement**
The authors declare that they have no conflict of interest.

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
