## [Review Process File · EMBO Molecular Medicine]

SARS-CoV-2 spike protein enhances MAP4K3/GLK-induced ACE2 stability in COVID-19

Huai-Chia Chuang, Chia-Hsin Hsueh, Pu-Ming Hsu, Rou-Huei Huang, Ching-Yi Tsai, Nai-Hsiang Chung, Yen-Hung Chow, and Tse-Hua Tan

DOI: [10.15252/emmm.202215904](https://doi.org/10.15252/emmm.202215904)

Corresponding author(s): Tse-Hua Tan (ttan@nhri.edu.tw), Huai-Chia Chuang (cinth@nhri.edu.tw), Yen-Hung Chow (choeyenh@nhri.edu.tw)

Review Timeline:

Submission Date:	17th Feb 22
Editorial Decision:	24th Mar 22
Revision Received:	22nd Jun 22
Editorial Decision:	4th Jul 22
Revision Received:	11th Jul 22
Accepted:	12th Jul 22

Editor: *Zeljko Durdevic*

Transaction Report:

24th Mar 2022

Dear Prof. Tan,

Thank you for the submission of your manuscript to EMBO Molecular Medicine. We have now received feedback from the three reviewers who agreed to evaluate your manuscript. As you will see from the reports below, the referees acknowledge the interest of the study but also raise important concerns that should be addressed in a major revision.

We would welcome the submission of a revised version within three months for further consideration. Please let us know if you require longer to complete the revision.

Please use this link to login to the manuscript system and submit your revision: <https://embomolmed.msubmit.net/cgi-bin/main.plex>

I look forward to receiving your revised manuscript.

Yours sincerely,

Zeljko Durdevic

**** Reviewer's comments ****

Referee #1 (Comments on Novelty/Model System for Author):

scRNA-seq and proteomics analyses using four COVID-19 patient cohorts including one from Taiwan.

The authors found that both protein levels and mRNA levels of GLK were increased in pseudovirus-infected HCC827 lung epithelial cells, as well as GLK levels increasing with the spike protein.

They also utilized UBR4 overexpression which enhanced Lys48-linked ubiquitination of ACE2 leading to proteasomal degradation of ACE2.

Referee #1 (Remarks for Author):

The current manuscript titled "SARS-CoV-2 spike protein enhances MAP4K3-induced ACE2 stability in COVID-19" describes whether GLK is involved in COVID-19 pathogenesis by scRNA-seq and proteomics analyses using four COVID-19 patient cohorts from Taiwan. Interestingly, the results suggest that GLK stabilizes ACE2 proteins in epithelial cells, enhancing SARS-CoV-2 infection susceptibility of epithelial cells. More specifically, the authors found that both protein levels and mRNA levels of GLK were increased in pseudovirus-infected HCC827 lung epithelial cells, as well as GLK levels increasing with the spike protein. They also found increased levels of the exosome marker CD63 in serum exosomes suggesting an enhancement of exosome release in COVID-19 patients. In situ proximity ligation assays (PLA) with ACE2 showed PLA signals in cells co-transfected with ACE2 and Flag-GLK, suggested a direct interaction between ACE2 and GLK. They also utilized UBR4 overexpression which enhanced Lys48-linked ubiquitination of ACE2 leading to proteasomal degradation of ACE2. Specificity controls included use of ACE2 (K26R), ACE2 (K112R), and ACE2 (K114R) and ACE2 (K94R) mutation resistant to UBR4-induced ACE2 degradation. Overall, this is an interesting and novel manuscript, however there are few minor concerns that include:

1. Have the authors performed any analysis on the exosomal contents? Are there any proteins related to the mitochondrial pathway? Any functional analysis of the exosomes on recipient cells?
2. Data in Figure 7 is very interesting, but somewhat overwhelming! Can the authors break up this figure into two and explain the results better in the result and discussion sections with more depth?

Referee #2 (Remarks for Author):

In this manuscript, Chuang et al found that SARS-CoV-2 infection, particularly the spike protein, can induce the GLK expression. In addition, GLK can induce release of exosome with ACE2 and stabilize ACE2 by reducing ubiquitination, thus promoting the SARS-CoV-2 infection. It is an interesting and novel pathway of SARS-CoV-2 pathogenesis, which will attract wide attentions from virologist. This pathway can also serve as a therapeutic target for COVID-19 treatment. This manuscript is well organized. The references are correctly cited. The methods are well described. However, some modifications should be made to further improve the solidity of the conclusion.

1. In figure 1C and D, and EV2, the figure legend indicates percentage while the figures show cell counts. The increase in GLK+ cell count cannot support that GLK expression was induced by SARS-CoV-2 infection since the total cell count in BLAF in SARS-CoV-2 patients are increased. So the percentage should be used here. Besides, the statistical analysis should use ANOVA instead of t-test since there are more than 2 groups. Same problem is also in figure 4I.
2. To test whether SARS-CoV-2 infection results in GLK overexpression in lung epithelial cells, the author used SARS-CoV-2

pseudovirus. A control pseudovirus with other viral surface protein should be added to see whether this GLK overexpression is specifically induced by SARS-CoV-2.

3. Similarly, in figure 3F, another pseudovirus should be used here to confirm whether the enhanced viral infection results from increased ACE2 level or other non-specific mechanism, such as enhanced endocytosis. Same problem is also in figure 4I.

4. In figure 4D, why the lower bands of ACE2 contains 3 bands.

5. The quality of Figure 5A is poor.

Minors:

1. In page 5, the author mentioned figure 1E and F, but no figure 1E and F in the figure section.

2. In methods section -- SARS-CoV-2 pseudovirus infection mouse model, the authors should indicate how many viruses should be added and when to measure in vivo fluorescence after pseudovirus infection.

Referee #3 (Comments on Novelty/Model System for Author):

1. Statistics could be improved (see specific comments). I would like to see an added control in the Co-IPs.

2. This is a very interesting and complete description of a new mechanism.

3. Currently, it's hard to estimate the actual impact the full mechanism, as described by the authors, has in COVID-19. Testing GLK-inhibition may reveal a potential new treatment route.

4. I don't see any issues here.

Referee #3 (Remarks for Author):

In their manuscript, Chuang et al. investigate the role of MAP4K3/GLK in a potential feedback cycle during SARS-CoV-2 infection. They find that

1. The SARS-CoV-2 spike protein induces the expression of GLK

2. GLK is linked to exosome numbers

3. These exosomes carry ACE2, the main SARS-CoV-2 entry receptor

4. The exosomes can transfer ACE2 to other cells

5. ACE2 protein levels are increased by GLK

6. This increase is mediated by direct interaction and phosphorylation of ACE2, in turn preventing ubiquitinylation and thus proteasomal degradation

7. These mechanisms can be observed in an ACE2-humanized mouse model system

From this they deduce that the S-protein sensitizes its host organism to infection by indirectly increasing the number of ACE2-carrying cells through the aforementioned mechanisms. I find the manuscript quite exciting and the individual mechanisms described to be plausible. One aspect I'm not fully convinced of at this point is whether the proposed mechanism in its entirety plays a significant role in COVID-19: while it is possible that GLK induction was the cause of increased exosome numbers in COVID-19 patients (and mice), the spike in exosomes could simply have been due to tissue damage during viral infection. This does not at all invalidate the author's findings, as the increase of ACE2 protein levels would still be relevant and ACE2 ends up in exosomes with or without GLK in this scenario. However, it would limit the role of GLK in the process and make it appear as a less interesting target for potential interventions. Thus, it would be interesting to see how much of the exosome release is due to GLK expression, which could be checked in the mouse model by comparing amounts of exosomes in the presence/absence of GLK inhibition. In case the authors do not want to follow up on this, I would at least like to see this point discussed. Other than this, I only have comments that should be straightforward to address:

- The title of the manuscript uses the name MAP4K3, whereas the main text generally uses GLK. This should be harmonized.
- Figure 1C: it is rather unlikely, that the requirements for a t-test are satisfied here. The variability in COVID-19 patients is likely to be increased by factors such as within-group variability of disease course, time of sampling, etc. Thus, homogeneity of variance is rather unlikely. Furthermore, while it won't massively affect the results here, in principle a correction for multiple testing would be appropriate. The authors could consider switching to Dunn's test or a similar, non-parametric multi-comparison test (see below).
- Comparisons based on absolute cell counts alone are not very meaningful in (clinical) scRNA-Seq data, as samples may not have even cell numbers (e.g. with Chua et al., the controls have lower cell numbers). Also, the percentage of positive cells strongly depends on the sequencing depth which in turn may be affected by cell quality (especially percentage of reads in cells). Still, comparing percentage of positive cells or splitting the population and running tools such as scCoda would be more robust.
- Fig. 1 E/F are missing
- It's unclear what „simple“ epithelial cells are supposed to be. Since ciliated cells are listed separately, would this refer to any epithelial cell type that's not ciliated (i.e. basal, goblet, club, ionocyte, FOXN4+, keratinocyte). In that case, is there a specific reason for this split? Basal, goblet, club, and ciliated cells belong to the same lineage, whereas the other cell types do not.
- Figure 3C: this roughly looks like there's distinct populations with respect to ACE2 levels within the COVID-19 cases, with patients 4, 7, and possibly 2 showing levels comparable to the healthy controls, at least when comparing to the total exosome number. Would the authors have any information regarding these patients, e.g. the severity of symptoms or time from onset of

symptoms?

- Figure 3D: the authors showed previously that GLK can increase the number of exosomes. If exosome production was mediated via some other mechanism (e.g. cell damage), the results could be explained by a lack of exosomes, unless similar amounts of exosomes were used per group.
- Figure 4B: could the authors demonstrate the absence of unspecific binding by having a +Myc, -FLAG control? Same goes for other Co-IPs.
- Figure 4G: could the authors provide an estimate of the half-life? The increase in baseline levels makes this a bit hard to determine visually
- Figure 4H: the addition of GLK does not seem to affect the ACE2 level here. Could the authors add some details about the timing (transfection to measurement) here? Was this shorter than for other experiments?
- Figure 6G: according to the labels in this panel, there is an ACE2-FLAG band in a sample where no FLAG-tagged protein should be present. Could the authors carefully check the labels here?
- Some antibody clones/catalog numbers appear to be missing (CD*)
- I did not get a hit for the ACE2 primers listed using either UCSC Genome Browser's in silico PCR or Primer Blast (against genome or transcriptome). Are the sequences correct?

Reply to Reviewer #1:

The current manuscript titled "SARS-CoV-2 spike protein enhances MAP4K3-induced ACE2 stability in COVID-19" describes whether GLK is involved in COVID-19 pathogenesis by scRNA-seq and proteomics analyses using four COVID-19 patient cohorts from Taiwan. Interestingly, the results suggest that GLK stabilizes ACE2 proteins in epithelial cells, enhancing SARS-CoV-2 infection susceptibility of epithelial cells. More specifically, the authors found that both protein levels and mRNA levels of GLK were increased in pseudovirus-infected HCC827 lung epithelial cells, as well as GLK levels increasing with the spike protein. They also found increased levels of the exosome marker CD63 in serum exosomes suggesting an enhancement of exosome release in COVID-19 patients. In situ proximity ligation assays (PLA) with ACE2 showed PLA signals in cells co-transfected with ACE2 and Flag-GLK, suggested a direct interaction between ACE2 and GLK. They also utilized UBR4 overexpression which enhanced Lys48-linked ubiquitination of ACE2 leading to proteasomal degradation of ACE2. Specificity controls included use of ACE2 (K26R), ACE2 (K112R), and ACE2 (K114R) and ACE2 (K94R) mutation resistant to UBR4-induced ACE2 degradation. Overall, this is an interesting and novel manuscript, however there are few minor concerns that include:

Reply: We thank the encouraging and constructive comments from Reviewer #1.

Comment 1. *Have the authors performed any analysis on the exosomal contents? Are there any proteins related to the mitochondrial pathway? Any functional analysis of the exosomes on recipient cells?*

Reply: We have listed serum exosomal proteins identified from COVID-19 patients and healthy controls (Source Data). The Cellular Component Ontology analysis shows that COVID-19 patient-enriched exosomal proteins mainly belong to membrane proteins, lysosomal proteins, and mitochondria proteins, while healthy control-enriched exosomal proteins mainly belong to cytosolic proteins and nuclear proteins (Appendix Fig S5). Moreover, KEGG pathway analysis shows that COVID-19 patient-enriched exosomal proteins may be involved in multiple metabolic pathways (Appendix Fig S6; page 7, line 16 to page 8, line 4). In addition, our results showed that GLK-induced ACE2-containing exosomes facilitated SARS-CoV-2 infection (Fig 3F, 7G, and 7H).

Comment 2. *Data in Figure 7 is very interesting, but somewhat overwhelming! Can the authors break up this figure into two and explain the results better in the result and discussion sections with more depth?*

Reply: Due to the limitation of the Figure number, we could not break up Figure 7. Nevertheless, we have re-organized Figure 7 (page 17, line 16 to page 18, line 9) and added subtitle to the result section (page 18, lines 10-11).

Reply to Reviewer #2:

In this manuscript, Chuang et al found that SARS-CoV-2 infection, particularly the spike protein, can induce the GLK expression. In addition, GLK can induce release of exosome with ACE2 and stabilize ACE2 by reducing ubiquitination, thus promoting the SARS-CoV-2 infection. It is an interesting and novel pathway of SARS-CoV-2 pathogenesis, which will attract wide attentions from virologist. This pathway can also serve as a therapeutic target for COVID-19 treatment.

This manuscript is well organized. The references are correctly cited. The methods are well described. However, some modifications should be made to further improve the solidarity of the conclusion.

Reply: We appreciate the encouraging and constructive comments from Reviewer #2.

Comment 1. *In figure 1C and D, and EV2, the figure legend indicates percentage while the figures show cell counts. The increase in GLK+ cell count cannot support that GLK expression was induced by SARS-CoV-2 infection since the total cell count in BLAF in SARS-CoV-2 patients are increased. So the percentage should be used here. Besides, the statistical analysis should use ANOVA instead of t-test since there are more than 2 groups. Same problem is also in figure 4I.*

Reply: We appreciate the constructive suggestion from Reviewer #2. Per Reviewer's comment, we have changed the data from cell count to the percentage (Figure 1C and 1D) accordingly. Moreover, we have performed the statistical analysis of Figure 1C, 1D, 3F, 4I, and 4J using ANOVA instead of *t-test* analysis accordingly (page 35, line 4; page 38, line 4; page 40, line 4).

Comment 2. *To test whether SARS-CoV-2 infection results in GLK overexpression in lung epithelial cells, the author used SARS-CoV-2 pseudovirus. A control pseudovirus with other viral surface protein should be added to see whether this GLK overexpression is specifically induced by SARS-CoV-2.*

Reply: Per Reviewer's comment, we have added vesicular stomatitis virus-G (VSV-G) pseudotyped lentivirus as a control pseudovirus and found that GLK levels were induced by SARS-CoV-2 but not VSV-G (Appendix Fig S4; page 5, lines 18-19).

Comment 3. *Similarly, in figure 3F, another pseudovirus should be used here to confirm whether the enhanced viral infection results from increased ACE2 level or other non-specific mechanism, such as enhanced endocytosis. Same problem is also in figure 4I.*

Reply: Per Reviewer's comment, we have treated GLK-overexpressing HCC827 epithelial cells with SARS-CoV-2 and VSV-G pseudovirus and found that "GLK overexpression did not enhance infection of VSV-G pseudovirus (Fig 4J)" (page 12, lines 12-13).

Comment 4. *In figure 4D, why the lower bands of ACE2 contains 3 bands.*

Reply: We have replaced Fig 4D with a clear figure.

Comment 5. *The quality of Figure 5A is poor.*

Reply: Per Reviewer's comment, we have replaced Fig 5A with a high-quality figure.

Minors:

Comment 1. *In page 5, the author mentioned figure 1E and F, but no figure 1E and F in the figure section?*

Reply: We have provided the data for Fig 1E and 1F, which were mistakenly placed in Expanded View Figures.

Comment 2. *In methods section -- SARS-CoV-2 pseudovirus infection mouse model, the authors should indicate how many viruses should be added and when to measure in vivo fluorescence after pseudovirus infection.*

Reply: Per Reviewer's comment, we have added the virus titer and measurement time point to the SARS-CoV-2 pseudovirus infection mouse model section in Materials and Methods (page 25, lines 7 and 10).

Reply to Reviewer #3:

Comments on Novelty/Model System for Author

- 1. Statistics could be improved (see specific comments). I would like to see an added control in the Co-IPs.*
- 2. This is a very interesting and complete description of a new mechanism.*
- 3. Currently, it's hard to estimate the actual impact the full mechanism, as described by the authors, has in COVID-19. Testing GLK-inhibition may reveal a potential new treatment route.*
- 4. I don't see any issues here.*

Reply: Per Reviewer's comment, we have improved statistical analyses (see below) and have added a control in the co-immunoprecipitation experiments (Fig 4B and 4H). Furthermore, we have added data of GLK inhibition in Fig EV5 (page 19, lines 3-15).

Remarks for Author

In their manuscript, Chuang et al. investigate the role of MAP4K3/GLK in a potential feedback cycle during SARS-CoV-2 infection. They find that

- 1. The SARS-CoV-2 spike protein induces the expression of GLK*
- 2. GLK is linked to exosome numbers*
- 3. These exosomes carry ACE2, the main SARS-CoV-2 entry receptor*

4. *The exosomes can transfer ACE2 to other cells*
5. *ACE2 protein levels are increased by GLK*
6. *This increase is mediated by direct interaction and phosphorylation of ACE2, in turn preventing ubiquitinylation and thus proteasomal degradation*
7. *These mechanisms can be observed in an ACE2-humanized mouse model system*

From this they deduce that the S-protein sensitizes its host organism to infection by indirectly increasing the number of ACE2-carrying cells through the aforementioned mechanisms. I find the manuscript quite exciting and the individual mechanisms described to be plausible. One aspect I'm not fully convinced of at this point is whether the proposed mechanism in its entirety plays a significant role in COVID-19: while it is possible that GLK induction was the cause of increased exosome numbers in COVID-19 patients (and mice), the spike in exosomes could simply have been due to tissue damage during viral infection. This does not at all invalidate the author's findings, as the increase of ACE2 protein levels would still be relevant and ACE2 ends up in exosomes with or without GLK in this scenario. However, it would limit the role of GLK in the process and make it appear as a less interesting target for potential interventions. Thus, it would be interesting to see how much of the exosome release is due to GLK expression, which could be checked in the mouse model by comparing amounts of exosomes in the presence/absence of GLK inhibition. In case the authors do not want to follow up on this, I would at least like to see this point discussed.

Reply: We thank the encouraging and constructive comments from Reviewer #3. Per Reviewer's comment #7, we have added the potential role of GLK in the enhancement of exosome release to the discussion section (page 9, lines 17-19; page 21, lines 3-5).

Other than this, I only have comments that should be straightforward to address:

Comment 1. *The title of the manuscript uses the name MAP4K3, whereas the main text generally uses GLK. This should be harmonized.*

Reply: Per Reviewer's comment, we have added "GLK" to the title (page 1, line 1).

Comment 2. *Figure 1C: it is rather unlikely, that the requirements for a t-test are satisfied here. The variability in COVID-19 patients is likely to be increased by factors such as within-group variability of disease course, time of sampling, etc. Thus, homogeneity of variance is rather unlikely. Furthermore, while it won't massively affect the results here, in principle a correction for multiple testing would be appropriate. The authors could consider switching to Dunn's test or a similar, non-parametric multi-comparison test (see below).*

Reply: Per Reviewer #3 (and also Reviewer #2)'s comments, we have performed the statistical analysis of Fig 1C, 1D using ANOVA instead of *t-test* analysis (page 35, line 4).

Comment 3. *Comparisons based on absolute cell counts alone are not very meaningful in (clinical) scRNA-Seq data, as samples may not have even cell numbers (e.g. with Chua et al., the controls have lower cell numbers). Also, the percentage of positive cells strongly depends on the sequencing depth which in turn may be affected by cell quality (especially percentage of reads in cells). Still, comparing percentage of positive cells or splitting the population and running tools such as scCoda would be more robust.*

Reply: Per Reviewer's comment, we have changed the data from cell count to the percentage (Figure 1C and 1D) accordingly (page 5, lines 2-5). Moreover, we have provided GLK levels in epithelial subgroups of individuals from Cohort #1 and Cohort #2; these results were statistically analyzed by *Kruskal-Wallis* analysis (Fig 1E, 1F and Appendix Fig S3C, S3D; page 5, lines 5-7 and 13-14).

Comment 4. *Fig. 1 E/F are missing.*

Reply: We have provided the data for Fig 1E and 1F, which were mistakenly placed in Expanded View Figures.

Comment 5. *It's unclear what „simple" epithelial cells are supposed to be. Since ciliated cells are listed separately, would this refer to any epithelial cell type that's not ciliated (i.e. basal, goblet, club, ionocyte, FOYN4+, keratinocyte). In that case, is there a specific reason for this split? Basal, goblet, club, and ciliated cells belong to the same lineage, whereas the other cell types do not.*

Reply: Per Reviewer's comment, we have changed "simple epithelial cells" to "KRT18⁺ epithelial cells" (page 5, line 2).

Comment 6. *Figure 3C: this roughly looks like there's distinct populations with respect to ACE2 levels within the COVID-19 cases, with patients 4, 7, and possibly 2 showing levels comparable to the healthy controls, at least when comparing to the total exosome number. Would the authors have any information regarding these patients, e.g. the severity of symptoms or time from onset of symptoms?*

Reply: Per Reviewer's comment, we have added the day from onset of COVID-19 patients and the treatment of oxygen therapy to Fig 3C (page 37, lines 4-5).

Comment 7. *Figure 3D: the authors showed previously that GLK can increase the number of exosomes. If exosome production was mediated via some other mechanism (e.g. cell damage), the results could be explained by a lack of exosomes, unless similar amounts of exosomes were used per group.*

Reply: For Fig 3D, we have previously demonstrated that GLK overexpression in epithelial

cells does not cause cell damage/death (Chuang *et al.*, 2019a), and there is no SARS-CoV-2 infection in Fig 3D. Nevertheless, we have added the sentence “we cannot rule out the possibility that SARS-CoV-2 infection may induce epithelial cell damage in COVID-19 patients, resulting in enhancement of exosome release” to the discussion section (page 21, lines 3-5).

Comment 8. *Figure 4B: could the authors demonstrate the absence of unspecific binding by having a +Myc, -FLAG control? Same goes for other Co-IPs.*

Reply: Per Reviewer’s comment, we have added a +Myc, -FLAG control in Fig 4B. We have also added a negative control in Fig 4H.

Comment 9. *Figure 4G: could the authors provide an estimate of the half-life? The increase in baseline levels makes this a bit hard to determine visually.*

Reply: Per Reviewer’s comment, we have added the estimated half-life of ACE2 protein to the sentence that “GLK overexpression prolonged ACE2 protein half-life (estimated half-life: 39.1 h to 157.5 h) in HEK293T cells” (page 12, line 5).

Comment 10. *Figure 4H: the addition of GLK does not seem to affect the ACE2 level here. Could the authors add some details about the timing (transfection to measurement) here? Was this shorter than for other experiments?*

Reply: The ratio of GLK to ACE2 was 1 in the original Fig 4H. In this revised manuscript, we have re-performed the experiment for Fig 4H under a condition that the ratio of GLK to ACE2 was 2.

Comment 11. *Figure 6G: according to the labels in this panels, there is an ACE2-FLAG band in a sample where no FLAG-tagged protein should be present. Could the authors carefully check the labels here?*

Reply: We have carefully checked the labels of Fig 6G and other Figures in this manuscript; we have corrected the typo error in Fig 6G.

Comment 12. *Some antibody clones/catalog numbers appear to be missing (CD*)*

Reply: We have added the clone number of anti-CD9 antibody to the Materials and Methods section (page 26, lines 19-20).

Comment 13. *I did not get a hit for the ACE2 primers listed using either UCSC Genome Browser's in silico PCR or Primer Blast (against genome or transcriptome). Are the sequences correct?*

Reply: We would like to clarify that in order to optimize the translation of human ACE2 mRNA

in mice, the sequences of hACE2 transgenic mice were modified using murine-optimized codons; the codon-optimized ACE2 cDNA sequences have been provided in Source Data 2 (page 24, line 21 and page 25, line 2).

4th Jul 2022

Dear Prof. Tan,

Thank you for the submission of your revised manuscript to EMBO Molecular Medicine. I am pleased to inform you that we will be able to accept your manuscript pending the following final amendments:

- 1) Please address the referee's comment by performing the appropriate statistical test.
- 2) In the main manuscript file, please do the following:
 - In M&M, provide the antibody dilutions that were used for each antibody.
 - In M&M, statistical paragraph should reflect all information that you have filled in the Authors Checklist, especially regarding randomization, blinding, replication.
 - In M&M, include a statement that informed consent was obtained from all human subjects and that, in addition to the WMA Declaration of Helsinki, the experiments conformed to the principles set out in the Department of Health and Human Services Belmont Report.
- 3) Authors Checklist: Please select "Yes" or "Not Applicable" in the column "Information included in the manuscript".
- 4) For more information: Please remove corresponding author's e-mail address. This space should be used to list relevant web links for further consultation by our readers. Could you identify some relevant ones and provide such information as well? Some examples are patient associations, relevant databases, OMIM/proteins/genes links, author's websites, etc...
- 5) Press release: Please inform us as soon as possible and latest at the time of submission of the revised manuscript if you plan a press release for your article so that our publisher could coordinate publication accordingly.
- 6) Please be aware that we use a unique publishing workflow for COVID-19 papers: a non-typeset PDF of the accepted manuscript is published as "Just Accepted" on our website. With respect to a possible press release, we have the option to not post the "Just Accepted" version if you prefer to wait with the press release for the typeset version. Please let us know whether you agree to publication of a "Just accepted" version or you prefer to wait for the typeset version.
- 7) As part of the EMBO Publications transparent editorial process initiative (see our Editorial at <http://embomolmed.embopress.org/content/2/9/329>), EMBO Molecular Medicine will publish online a Review Process File (RPF) to accompany accepted manuscripts. This file will be published in conjunction with your paper and will include the anonymous referee reports, your point-by-point response and all pertinent correspondence relating to the manuscript. Let us know whether you agree with the publication of the RPF and as here, if you want to remove or not any figures from it prior to publication. Please note that the Authors checklist will be published at the end of the RPF.
- 8) Please provide a point-by-point letter INCLUDING my comments as well as the reviewer's reports and your detailed responses (as Word file).

I look forward to reading a new revised version of your manuscript as soon as possible.

Yours sincerely,

Zeljko Durdevic

***** Reviewer's comments *****

Referee #3 (Comments on Novelty/Model System for Author):

The manuscript puts forward novel and exciting ideas, shows the required rigor, and highlights potential future avenues for understanding and treating COVID-19 better. The medical impact is not immediate, as the findings are not actionable at the moment, but may very well be in the future.

Referee #3 (Remarks for Author):

The authors have addressed my concerns comprehensively and alleviated reservations I had about some results. As requested by the editor, I also checked the response to reviewer 2 and am confident that the concerns were adequately addressed. I recommend the manuscript for publication, but have one small request: Reviewer 2 and I commented on the statistics for Figures 1C and D and 4I. Switching to ANOVA is an improvement, but could the authors add a post-hoc test (Dunn's or Dunnett's test) on top to show differences between groups? The differences are clear, so this will not require re-review.

Reply to Reviewer #3:***Comments on Novelty/Model System for Author***

The manuscript puts forward novel and exciting ideas, shows the required rigor, and highlights potential future avenues for undersanding and treating COVID-19 better. The medical impact is not immediate, as the findings are not actionable at the moment, but may very well be in the future.

Reply: We thank the encouraging and constructive comments from Reviewer #3.

Remarks for Author

The authors have addressed my concerns comprehensively and alleviated reservations I had about some results. As requested by the editor, I also checked the response to reviewer 2 and am confident that the concerns were adequately addressed. I recommend the manuscript for publication, but have one small request: Reviewer 2 and I commented on the statistics for Figures 1C and D and 4I. Switching to ANOVA is an improvement, but could the authors add a post-hoc test (Dunn's or Dunnett's test) on top to show differences between groups? The differences are clear, so this will not require re-review.

Reply: Per Reviewer's comment, we have added a *post-hoc test (Dunnett's test)* to Figure 1C, 1D, and 4I.

We are pleased to inform you that your manuscript is accepted for publication and is now being sent to our publisher to be included in the next available issue of EMBO Molecular Medicine.